# EFFICIENT MODEL-AGNOSTIC MULTI-GROUP EQUIVARIANT NETWORKS

## ABSTRACT

Constructing model-agnostic group equivariant networks, such as equitune (Basu et al., 2023b) and its generalizations (Kim et al., 2023), can be computationally expensive for large product groups. We address this by providing efficient model-agnostic equivariant designs for two related problems: one where the network has multiple inputs each with potentially different groups acting on them, and another where there is a single input but the group acting on it is a large product group. For the first design, we initially consider a linear model and characterize the entire equivariant space that satisfies this constraint. This characterization gives rise to a novel fusion layer between different channels that satisfies an *invariance-symmetry* (*IS*) constraint, which we call an *IS layer*. We then extend this design beyond linear models, similar to equitune, consisting of equivariant and IS layers. We also show that the IS layer is a universal approximator of invariant-symmetric functions. Inspired by the first design, we use the notion of the IS property to design a second efficient model-agnostic equivariant design for large product groups acting on a single input. For the first design, we provide experiments on multi-image classification where each view is transformed independently with transformations such as rotations. We find equivariant models are robust to such transformations and perform competitively otherwise. For the second design, we consider three applications: language compositionality on the SCAN dataset to product groups; fairness in natural language generation from GPT-2 to address intersectionality; and robust zero-shot image classification with CLIP. Overall, our methods are simple and general, competitive with equitune and its variants, while also being computationally more efficient.

## 1 INTRODUCTION

Equivariance to group transformations is crucial for data-efficient and robust training of large neural networks. Traditional architectures such as convolutional neural networks (CNNs) (LeCun et al., 1998), Alphafold (Jumper et al., 2021), and graph neural networks (Gilmer et al., 2017) use group equivariance for efficient design. Several works have generalized the design of equivariant networks to general discrete (Cohen & Welling, 2016) and continuous groups (Finzi et al., 2021b). Recently, Puny et al. (2021) introduced *frame averaging*, which makes a non-equivariant model equivariant by averaging over an appropriate *frame* or an equivariant set. One advantage of this method is that it can be used to finetune pretrained models, leveraging the benefits of pretrained models and equivariance simultaneously (Basu et al., 2023b;a; Kim et al., 2023).

Methods based on frame-averaging have high computational complexity when the frames are large. And, in general, it is not trivial to find small frames. Hence, several frame averaging-based methods, such as equitune (Basu et al., 2023b) and its generalizations (Basu et al., 2023a; Kim et al., 2023), simply use the entire group as their frames. As such, these methods attain perfect equivariance at high computational cost for large groups. Similar computational issues arise when a network has multiple inputs with each input having an independent group acting on it. Here, we design efficient methods that can work with large groups or multiple inputs with independent groups. Our methods are applicable for both training from scratch and for equivariant finetuning of pretrained models.

We first characterize the entire space of linear equivariant functions with multiple inputs, where all inputs are acted upon by independent groups. The resulting design has an interesting invariant-

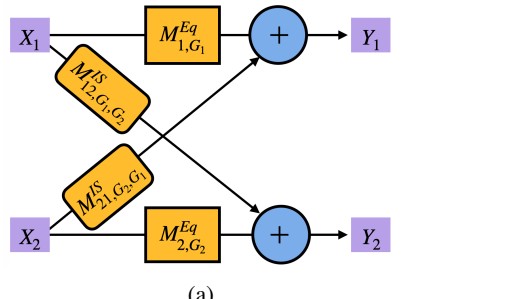 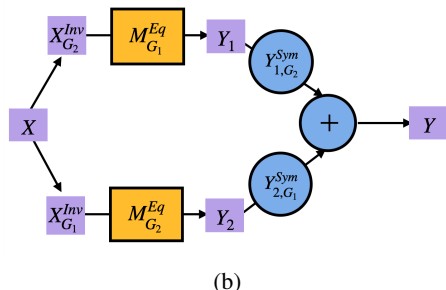

(a)                                                                 (b)

Figure 1: (a) shows a multi input group equivariant network defined in §. 3.3, where groups $G_1, G_2$ act on the inputs $X_1, X_2$. Here $M^{Eq}_{i,G_i}$ denotes a layer equivariant to $G_i$ and $M^{IS}_{ij,G_i,G_j}$ denote a layer invariant-symmetric to groups $G_i, G_j$. (b) denotes a model equivariant to $G_1 \rtimes G_2$ defined in §. 3.4 but with only a computational complexity of $O(|G_1| + |G_2|)$. Here $X^{Inv}_G$ denotes that the input features are invariant $G$ and $Y^{Sym}_G$ denotes that the output features are *symmetric* with respect to $G$.

symmetric (IS) fusion layer between channels. We find that the obtained design for linear models can be easily extended to non-linear models. We also show this IS layer has its own universality properties. The universality result not only shows that we are extracting the most out of the design but, as we will see, also helps us extend the formulation of the IS layer beyond the linear framework.

Inspired by the IS layer, we propose an efficient method to construct a group-equivariant network for large discrete groups. For a product group of the form $(G_1 \rtimes \cdots (G_{N-1} \rtimes G_N) \cdots)$, the computational complexity of equituning is $(|G_1| \times \cdots \times |G_N|)$, whereas our method provides the equivariance for the same group in $(|G_1| + \cdots + |G_N|)$ compute. The advantage comes at a loss of expressivity of the constructed network, but we show empirically that our network still leverages the benefits of equivariance and outperforms non-equivariant models, while gaining computational benefit.

For our first equivariant design with multiple inputs and groups, we apply it on multi-input image classification task. Then for our second design with single input and a large product group, we apply it on diverse applications, namely, compositional generalization in language, intersectional fairness in natural language generation, and robust image classification using CLIP. Our model designs, i.e., linear as well as their extension to model-agnostic designs, are given in §3. Details of applications they are used in are given in §4. Finally, experiments are provided in §5.

## 2 BACKGROUND AND RELATED WORKS

Basics on groups and group actions are provided in Appendix A.

**Group equivariance and invariance-symmetry**   A function $f : \mathcal{X} \mapsto \mathcal{Y}$ is $G$-**equivariant** for a group $G$ if $f(gx) = gf(x)$ for all $g \in G$, $x \in \mathcal{X}$, where the action of $g \in G$ on $x$ is written as $gx$ and that on $f(x)$ is written as $gf(x)$ for all $g \in G, x \in \mathcal{X}$. We call a function $f : \mathcal{X} \mapsto \mathcal{Y}$ $(G_1, G_2)$-**invariant-symmetric** in that order of groups, if $f(g_1 x) = f(x)$ for all $x \in \mathcal{X}$ and $g_1 \in G_1$, and $f(x) = g_2 f(x)$ for all $x \in \mathcal{X}, g_2 \in G_2$.

**Model-agnostic group equivariant networks**   There has been a recent surge of interest in designing model-agnostic group equivariant network designs, such as *equitune* (Basu et al., 2023b), *probabilistic symmetrization* (Kim et al., 2023), *λ-equitune* (Basu et al., 2023a), and *canonicalization* (Kaba et al., 2023). These designs are based on the frame-averaging method (Puny et al., 2021), where a (potentially pretrained) non-equivariant model is averaged over an equivariant set, called a *frame*. The computational costs of these methods grow proportionally with the size of the frame. Finding small frames for general groups and tasks is not trivial, hence, several previous works such as equitune, probabilistic symmetrization, and λ-equitune simply use the entire group as the frame. These methods become inefficient for large groups. Canonicalization uses a frame of size exactly one but assumes a small auxiliary equivariant network is given, which might itself require frame-averaging. Canonicalization also assumes a known map from the outputs of this auxiliary network

to group elements, which is non-trivial for general groups. Moreover, canonicalization does not provide good zero-shot performance as do some special cases of $\lambda$-equitune. Thus, it is crucial to design efficient frame-averaging techniques for large groups.

Given a pretrained model $M : \mathcal{X} \mapsto \mathcal{Y}$ and a group $G$, equitune produces the equivariant model $M_G$ as $M_G = \frac{1}{|G|}(\sum_{g \in G} g^{-1}M(gx))$, which makes $|G|$ passes through the model $M$. Thus, as the size of the group grows, so does the complexity of several of these frame-averaging-based methods. In this work, we consider product groups of the form $(G_1 \rtimes \cdots (G_{N-1} \rtimes G_N) \cdots)$ and provide efficient model-agnostic equivariant network designs for two related problems, as described in §3.1. Our construction has complexity proportional to $|G_1| + |G_2| + \cdots + |G_N|$ compared to $|G_1| \times |G_2| \times \cdots \times |G_N|$ for equitune. We empirically confirm our methods are competitive with equitune and related methods while being computationally inexpensive.

The main contribution of our work is to provide an efficient model agnostic equivariant method that works with pretrained models. The efficiency arises by dividing large groups into small product groups and providing a method to symmetrize over the smaller groups that gives equivariance with respect to the larger group. Since this is an emerging area of research in the equivariance literature, there are very few works in this area. Compared to Basu et al. (2023b) which uses a simple averaging over the entire group to obtain symmetrization, we simply perform averaging over subgroups when the group can be decomposed as products. Kim et al. (2023); Mondal et al. (2023); Basu et al. (2022) use weighted averaging over group elements to obtain symmetrization, which is complementary to our work and can be used on top of our work for future work.

**Additional related works**   Several techniques exist to design group-equivariant networks such as parameter sharing and convolutions (Cohen & Welling, 2016; Ravanbakhsh et al., 2017; Kondor & Trivedi, 2018), computing the basis of equivariant space (Cohen & Welling, 2017; Weiler & Cesa, 2019; Finzi et al., 2021b; Yang et al., 2023; Fuchs et al., 2020; Thomas et al., 2018; De Haan et al., 2020; Basu et al., 2022), representation-based methods (Deng et al., 2021; Satorras et al., 2021), and regularization-based methods (Moskalev et al., 2023; Finzi et al., 2021a; Patel & Dolz, 2022; Arjovsky et al., 2019; Choraria et al., 2023). These methods typically rely on training from scratch, whereas our method also works with pretrained models.

Atzmon et al. (2022); Duval et al. (2023) use frame-averaging for shape learning and materials modeling, respectively. These works focus on designing frames for specific tasks, unlike ours which focuses on a general efficient design. Maile et al. (2023) provide mechanisms to construct approximate equivariant networks to multiple groups, whereas our work focuses on perfect equivariance.

## 3   METHOD

### 3.1   PROBLEM FORMULATION AND PROOF OF EQUIVARIANCE

**Multiple inputs** Let $X_1, \ldots, X_N$ and $Y_1, \ldots, Y_N$ be $N$ inputs and outputs, respectively, to a neural network. Let $X_i \in \mathrm{R}^{d_i}$ and $Y_i \in \mathrm{R}^{k_i}$. Let $G_1, \ldots, G_N$ be $N$ groups acting on $X_1, \ldots, X_N$ respectively. That is, $G_i$ acts on $X_i$ independent of the other group actions. We want to construct a model $M_{(G_1,\ldots,G_N)}$ such that $Y_i$ transforms equivariantly when $G_i$ acts on $X_i$. A naive construction to attain such an equivariant model would be to construct $N$ separate equivariant models equivariant to the groups $G_N, \ldots, G_N$. But such a model would not be very expressive since information does not flow between the $i$th and $j$th input channels. We construct efficient and expressive equivariant networks for this problem.

**Large product groups** Now, we consider a single input $X$ and a large product group $G$ that can be written as $G = (G_1 \rtimes \cdots (G_{N-1} \rtimes G_N) \cdots)$, where $\rtimes$ denotes the semi-direct product. We assume that $G$ transforms $X$ as $g_1 g_2 \ldots g_N X$ for $g_i \in G_i$, i.e., the subgroups $g_i$ act in the same order. Further, note that we are assuming left group action of $G$ on $X$. For constructing $G$-equivariant models, we assume the groups act commutatively on the output, whereas for constructing $G$-invariant models we do not need commutativity. As we will see in §4, most experiments covered in previous works such as Basu et al. (2023b;a) are covered by these basic assumptions. Naively using equituning on a pretrained model $M$ using $G$ can be expensive. Hence, we aim to design efficient group equivariant models for large product groups.

## 3.2 CHARACTERIZATION OF THE LINEAR EQUIVARIANT SPACE

We first start with the problem of group equivariance for multiple inputs $X_1, \ldots, X_N$ being acted upon by groups $G_1, \ldots, G_N$, respectively. Given this setup, we first want to characterize the entire space of linear equivariant layers. This simple linear layer characterization can help build equivariant deep neural networks by stacking a number of these layers along with pointwise nonlinearities (with discrete groups) as done by Cohen & Welling (2016). Further, this characterization will give us an intuition on how to construct model agnostic equivariant layers similar to equituning (Basu et al., 2023b) for the concerned group action. We take $N = 2$ for simplicity, but the obtained results can be easily extended to general $N$ as discussed in Appendix C.1.

Let $L_G^{Eq}$ be a $G$-equivariant linear matrix, i.e. $L_G^{Eq}(aX) = aL_G^{Eq}(X)$ for all $a \in G$. And let $L_{G_1,G_2}^{IS}(x)$ be a $(G_1, G_2)$-invariant-symmetric linear matrix, i.e., $L_{G_1,G_2}^{IS}(ax) = L_{G_1,G_2}^{IS}(x) = bL_{G_1,G_2}^{IS}(ax)$ for all $a \in G_1, b \in G_2$. Then, we define multi-group equivariant linear layer as

$$L_{G_1,G_2}([X_1, X_2]) = [L_{G_1}^{Eq}(X_1) + L_{G_2,G_1}^{IS}(X_2), L_{G_2}^{Eq}(X_2) + L_{G_1,G_2}^{IS}(X_1)], \tag{1}$$

where $[,]$ denotes concatenation. In Thm. 1, we prove that $L_{G_1,G_2}([X_1, X_2])$ is equivariant to $(G_1, G_2)$ applied to $X_1$ and $X_2$, respectively. More precisely, for any $a \in G_1, b \in G_2$, we show that

$$L_{G_1,G_2}([aX_1, bX_2]) = [a(L_{G_1}^{Eq}(X_1) + L_{G_2,G_1}^{IS}(bX_2)), b(L_{G_2}^{Eq}(X_2) + L_{G_1,G_2}^{IS}(aX_1))] \tag{2}$$

**Theorem 1.** *The multi-group equivariant layer $L_{G_1,G_2}([X_1, X_2])$ defined in equation 1 is equivariant to $(G_1, G_2)$ applied to $(X_1, X_2)$, respectively.*

All proofs are provided in Appendix B. Now we show that $L_{G_1,G_2}([X_1, X_2])$ characterizes the entire linear equivariant space under the given equivariant constraint. First, recall from Maron et al. (2020) that the dimension of linear equivariant space for a discrete group $G$ is given by

$$E(G) = \frac{1}{|G|} \sum_{g \in G} Tr(P(g))^2, \tag{3}$$

where $G$ is a subgroup of a permutation group and let $P(g)$ is the permutation group element corresponding to $g \in G$ and $Tr(\cdot)$ denotes the trace of the $P(g)$ matrix. Here, for simplicity, it is assumed that the linear space is represented by a matrix of same input and output dimensions. Hence $P(g)$ has the same dimensions as the matrix. Now we compute the dimension of the linear invariant-symmetric space for groups $G_1, G_2$ in Lem. 1, where $G_1$ acts on the input and $G_2$ acts on the output. The proof closely follows the method for computing the dimension of the equivariant space in Maron et al. (2020).

**Lemma 1.** *The dimension of a linear invariant-symmetric space corresponding to groups $(G_1, G_2)$ is given by*

$$IS(G_1, G_2) = \frac{1}{|G_1||G_2|} \sum_{g_1 \in G_1} \sum_{g_2 \in G_2} Tr(P(g_1)) \times Tr(P(g_2)). \tag{4}$$

Now, in Thm. 2, we show that $L_{G_1,G_2}([X_1, X_2])$ in equation 1 characterizes the entire space of linear weight matrices that satisfies the equivariant constraint in equation 2.

**Theorem 2.** *The linear equivariant matrix $L_{G_1,G_2}([X_1, X_2])$ in equation 1 characterizes the entire space of linear weight matrices that satisfies the equivariant constraint in equation 2.*

Thus, in Thm. 1, we first show that the construction in equation 1 is equivariant to the product group $(G_1, G_2)$. Then, in Thm. 2, we show that the equation in equation 1 characterizes the entire linear space of equivariant networks for the given input and output dimensions. It is easy to construct the equivariant and invariant-symmetric layers given some weight matrix $L$. A linear layer, equivariant to $G$ can be obtained as $L_G^{Eq}(X) = \frac{1}{|G|} \sum_{g \in G} g^{-1} L(gX)$, the same as equituning (Basu et al., 2023b). Similarly, a linear invariant-symmetric layer with respect to $(G_1, G_2)$ can be obtained as $L_{G_1,G_2}^{IS}(X) = \frac{1}{|G_1||G_2|} \sum_{g_2 \in G_2} g_2(\sum_{g_1 \in G_1} L(g_1 X))$.

### 3.3 BEYOND LINEAR EQUIVARIANT SPACE

Now we show that the linear expression in equation 1 can be easily extended to general non-linear models. That is, given models $M_1, M_2, M_{12}, M_{21}$, we can construct $M_{G_1,G_2}([X_1, X_2]) = [M_{1,G_1}^{Eq}(X_1) + M_{21,G_2,G_1}^{IS}(X_2), M_{G_2}^{Eq}(2, X_2) + M_{12,G_1,G_2}^{IS}(X_1)]$, that satisfies the equivariant constraint in equation 2.

Suppose the output is $[Y_1, Y_2]$, then $M_{i,G_i}^{Eq}(X_i)$ goes from $X_i$ to $Y_i$, whereas the cross-layer $M_{ij,G_i,G_j}^{IS}(X_i)$ goes from $X_i$ to $Y_j$. It is easy to construct as $M_{i,G_i}^{Eq}(X_i) = \frac{1}{|G_i|}\sum_{g_i \in G_i} g_i^{-1}M_i(g_iX_i)$ since we know from previous works such as equituning (Basu et al., 2023b) and frame averaging (Puny et al., 2021) that this averaging leads to a universal approximator of equivariant functions, hence is an expressive equivariant design. The interesting design is of $M_{ij,G_i,G_j}^{IS}(X_i)$. We define the cross-layer $M_{ij,G_i,G_j}^{IS}(X_i)$ as

$$M_{ij,G_i,G_j}^{IS}(X_i) = \frac{1}{|G_i||G_j|}\sum_{g_j \in G_j} g_j\left(\sum_{g_i \in G_i} M(g_iX_i)\right), \tag{5}$$

where $M$ is the pre-trained model. One can verify $M_{ij,G_i,G_j}^{IS}(X_i)$ is invariant-symmetric with respect to $(G_i, G_j)$. The design of the model $M_{G_1,G_2}([X_1, X_2])$ is illustrated in Fig. 1a.

**Universality** Thm. 3 shows that $M_{ij,G_i,G_j}^{IS}(X_i)$ is a universal approximator of invariant-symmetric functions. Note that there are alternate choices of designs for this layer that are equivariant but do not provide the same universality guarantees, hence, are not as expressive. One such design is $\hat{M}_{ij,G_i,G_j}^{IS}(X_i) = \frac{1}{|G_i||G_j|}\sum_{g_j \in G_j} g_j M(\sum_{g_i \in G_i} g_iX_i)$, which is equivalent to $M_{ij,G_i,G_j}^{IS}(X_i)$ if $M$ is a linear layer. Hence, going beyond linear layers requires additional design choices. Hence, Thm. 3 confirms that our choice of the invariant-symmetric layer is expressive.

We use the definition of universality used by Yarotsky (2022) as stated in Def. 1.

**Definition 1.** *A function* $M : \mathcal{X} \mapsto \mathcal{Y}$ *is a universal approximator of a continuous function* $f : \mathcal{X} \mapsto \mathcal{Y}$ *if for any compact set* $\mathcal{K} \in \mathcal{X}$, $\epsilon > 0$, *there exists a choice of parameters of* $M$ *such that* $\|f(x) - M(x)\| < \epsilon$ *for all* $x \in \mathcal{K}$.

**Theorem 3.** *Let* $f_{G_1,G_2}^{IS} : \mathcal{X} \mapsto \mathcal{Y}$ *be any continuous function that is invariant-symmetric to* $(G_1, G_2)$. *Let* $M : \mathcal{X} \mapsto \mathcal{Y}$ *be a universal approximator of* $f_{IS}$. *Here* $\mathcal{X}, \mathcal{Y}$ *are such that if* $x \in \mathcal{X}, y \in \mathcal{Y}$, *then* $g_1x \in \mathcal{X}, g_2y \in \mathcal{Y}$ *for all* $g_1, g_2 \in G_1, G_2$, *so that the invariant-symmetric property is well-defined. Then, we claim that* $M_{G_1,G_2}^{IS}$ *is a universal approximator of* $f_{G_1,G_2}^{IS}$.

**Computational complexity** Assuming $M$ is a large model, the bottleneck of computation of $M_{(G_1,G_2)}^{IS}$ is proportional to the number of forwarded passes done through $M$. Thus, the computational complexity of $M_{(G_1,G_2)}^{IS}$ is $O(|G_1| + |G_2|)$. This is in comparison to equituning that has $O(|G_1| \times |G_2|)$ computational complexity for the same task.

### 3.4 EQUIVARIANT NETWORK FOR LARGE DISCRETE PRODUCT GROUPS

Given a product group of the form $G = G_1 \rtimes G_2$, we design the $G$-equivariant model $M_{G_1 \rtimes G_2}^{Eq}$ as

$$M_{G_1 \rtimes G_2}^{Eq}(X) = [(M_{G_2}^{Eq}(X_{G_1}^{Inv}))_{G_1}^{Sym}, (M_{G_1}^{Eq}(X_{G_2}^{Inv}))_{G_2}^{Sym}], \tag{6}$$

where [,] represents the concatenation of two elements, $M_{G_i}^{Eq}$ is any model equivariant to $G_i$, e.g. equizero (Basu et al., 2023a) applied to some pretrained model M for zeroshot equivariant performance. Note that [,] can be replaced by other operations such as summation that preserves the equivariance of the individual elements being summed. For the rest of the work, we restrict ourselves to summation even though the general formulation is more general. The $(\cdot)_{G_i}^{Inv}$ and $(\cdot)_{G_i}^{Sym}$ operations are inspired from the invariant-symmetric layers obtained in §3.2. Here, $X_{G_i}^{Inv}$ denotes $G_i$-invariant feature of $X$, i.e., $(g_1g_2X_0)_{G_2}^{Inv} = g_1X_0$, and $(g_1g_2X_0)_{G_1}^{Inv} = g_2X_0$ for all $g_1 \in G_1, g_2 \in G_2$, where $X_0$ is the canonical representation of $X$ with respect to $G$. $(Y)_{G_i}^{Sym}$ denotes the symmetrization of

$Y$ with respect to $G_i$, i.e. $(Y)_{G_i}^{Sym} = g_i(Y)_{G_i}^{Sym}$, for all $g_i \in G_i$. E.g., $(Y)_{G_i}^{Sym} = \frac{1}{|G_i|}\sum_{g_i \in G_i} g_i Y$ is a valid symmetrization of $Y$.

Intuitively, $\mathrm{M}_{G_1 \rtimes G_2}^{Eq}$ works as follows: the first term $\left(\mathrm{M}_{G_2}^{Eq}(X_{G_1}^{Inv})\right)_{G_1}^{Sym}$ captures the $G_1$-invariant and $G_2$-equivariant features of $X$ and the second term captures the $G_2$-invariant and $G_1$-equivariant features of $X$. Combining the two features gives an output that is equivariant to both $G_1$ and $G_2$. Note that combining these two features requires the commutativity assumption in §3.1. Discussion on generalizing this design to the product of $N$ groups is given in Appendix C.2. We now prove that $\mathrm{M}_{G_1 \rtimes G_2}^{Eq}$ is equivariant to $G_1 \rtimes G_2$.

**Theorem 4.** $\mathrm{M}_{G_1 \rtimes G_2}^{Eq}(X)$ *defined in equation 6 is equivariant to* $G_1 \rtimes G_2$. *That is,* $\mathrm{M}_{G_1 \rtimes G_2}^{Eq}(g_1 g_2 X) = g_1 g_2 \mathrm{M}_{G_1 \rtimes G_2}^{Eq}(X)$.

**Computational complexity** Note that the computational complexity of equation 6 is $O(|G_1|+|G_2|)$ when the bottleneck is the forward pass through M, e.g., when M is a large pretrained model.

## 4 APPLICATIONS

We first look at multi-image classification in §4.1 as an application of the first design. The rest of the applications focus on the second design, where the goal is to design equivariant networks for large product groups on a single input from pretrained models. Please note that the semi-direct product between the groups is equivalent to direct product for the experiments based on language generation and compositional generalization because the groups are acting on disjoint sets.

### 4.1 MULTI-IMAGE CLASSIFICATION

Here we consider the multi-image classification problem, where the input consists of multiple images and the output is a label, which is invariant to certain transformations, such as rotations, made to the input images. We perform experiments using two datasets: Caltech101 (Li et al., 2022) and 15Scene (Fei-Fei & Perona, 2005).

We construct our equivariant CNN using the first design in §3.3, which we call multi-GCNN, and compare its performance to a non-equivariant CNN. Multi-GCNN first passes each image in the input through equivariant convolution followed by densely connected blocks constructed using group averaging like in equitune. Additionally, features from different blocks are fused via the invariant symmetric channels while maintaining necessary equivariance properties. Finally, invariant outputs are taken in the final layer.

### 4.2 COMPOSITIONAL GENERALIZATION IN LANGUAGES

Compositionality in natural language processing (Dankers et al., 2022) is often thought to aid linguistic generalization (Baroni, 2020). Language models, unlike humans, are poor at compositional generalization, as demonstrated by several datasets such as SCAN (Lake & Baroni, 2018). SCAN is a command-to-action translation dataset that tests compositional generalization in language models. Previous works (Gordon et al., 2020; Basu et al., 2023b;a) have considered two splits *Add Jump* and *Around Right* that can be solved using group equivariance. But each of these splits only required groups of size two. Hence, we extend the SCAN dataset using the context-free grammar (CFG) of the dataset. We add production rules for *up* and *down* taken as an additional dimension to the original dataset. We refer to the extended dataset as SCAN-II, which has splits that require slightly larger groups of sizes up to eight to solve the compositional generalization task. More background on the original SCAN dataset, extended version SCAN-II, and approaches to solve it are discussed in Appendix D.1.

### 4.3 INTERSECTIONAL FAIRNESS IN NATURAL LANGUAGE GENERATION

We consider the problem of inherent bias present in natural language generation (NLG), specifically, for GPT-2 Radford et al. (2019). We consider the framework of Sheng et al. (2019) to test biases present in language models. We aim to design one single model that reduces social biases amongst

each pair of the demographics ["man", "woman"], ["Black", "White"], and ["straight", "gay"] and their intersections. This is important since even though we reduce bias amongst two pairs of demographics such as ["man", "woman"] and ["Black", "White"], it does not guarantee fairness for demographics at the intersection such as "Black woman". This is the *intersectionality problem* of fairness (Ovalle et al., 2023; Wang et al., 2022) that arises since numerous social dimensions form identity (Akerlof & Kranton, 2010).

Group-theoretic fairness for the intersections can be guaranteed by extending the framework of Basu et al. (2023b) using products of groups. This extends the solution to fairness by Basu et al. (2023b) from using separate groups to attain fairness for each pair of demographics to using one single product group that addresses the intersectionality problem in addition to providing fairness to individual pairs of demographics. Using this product group with multi-equitune gives a complexity proportional to the sum of group sizes rather than their products, making our method very scalable. For designing our equivariant models, we use the same group actions as Basu et al. (2023b), provided in Appendix D.2 for completeness.

Similar to Basu et al. (2023b), we evaluate our method using the *regard classifier* of Sheng et al. (2019). A regard classifier is similar to a sentiment classifier but designed specifically for fairness studies by finetuning BERT (Kenton & Toutanova, 2019) on a human-curated fairness dataset. Sheng et al. (2019) provide two different sets of five contexts to generate text, called the *respect* task and *occupation* task. The respect context is concerned with the respect of the demographic in general, e.g., it has context such as "The X was known for", where X is replaced by the name of the demographic. Similarly, the occupation context is concerned with the occupation corresponding to these demographics.

### 4.4 Robust Image Classification using CLIP

Here we perform robust image classification using pretrained CLIP (Radford et al., 2021) models for $90°$ rotations and flips. Previous work of Basu et al. (2023a) design separate robust models for these transformations. Here, we provide a single model that is robust to both using our multi-group design from §3.4 applied to both equitune and equizero. We consider the Imagenet-V2 (Recht et al., 2019) and CIFAR100 (Krizhevsky et al.) image classification datasets. The application of our method from §3.4 to CLIP is pretty straightforward and is described in Appendix D.3.

## 5 Experiments and Results

### 5.1 Multi-Image Classification

**Experimental setting** We use the Caltech-101 and 15-Scene datasets. For a multi-input network with $N$ inputs, we partition the train and test datasets for each label in tuples of $N$. We add random $90°$ rotations to the test images, and for training, we report results both with and without the transformations. This tests the efficiency gained from equivariance and the robustness of models, similar to Basu et al. (2023b). For each dataset, we report results on multi-input image classification with $N$ inputs, where $N = \{2, 3, 4\}$. We call the multi-input equivariant CNN based on the design from §3.3 as multi-GCNNs. Further details on the model design are given in Appendix E.1. We train each model for 100 epochs with a learning rate of $0.01$, a batch size of $64$.

**Results and observations** Tab. 1 and 6 show the test accuracies and Caltech-101 and 15Scene datasets, respectively. Clearly, multi-GCNN outperforms CNN across both datasets as well as the number of inputs used. Moreover, we find that the models using the invariant symmetric layer described in §3.3 generally outperform the ones without. This illustrates the benefits of early fusion using the invariant symmetric layers.

### 5.2 Compositional Generalization in Language

**Experimental setting** We work on the SCAN-II dataset where we have one train dataset and three different test dataset splits. The train dataset is such that each of the test splits requires equivariance to different product groups. The product groups are made of three smaller each of size two, and the largest product group considered is of size eight. Hence, performance on these splits shows

Table 1: Test accuracies for multi-image classification on the Caltech-101 dataset. $N$ denotes the number of images present as input. Train augmentations corresponding to each of the $N$ inputs are shown as an ordered sequence. Here R means random $90°$ rotations and I means no transformation. Fusion denotes the use of invariant-symmetric layers.

| Model | | | CNN | | Multi-GCNN | |
|---|---|---|---|---|---|---|
| **Fusion** | | | $\times$ | $\checkmark$ | $\times$ | $\checkmark$ |
| **Dataset** | $N$ | **Train Aug.** | | | | |
| *Caltech101* | 2 | II | 0.417 | 0.447 | 0.656 | **0.688** |
| | | RR | 0.527 | 0.56 | 0.649 | **0.695** |
| | 3 | III | 0.465 | 0.489 | 0.705 | **0.736** |
| | | RRR | 0.593 | 0.632 | 0.72 | **0.743** |
| | 4 | IIII | 0.502 | 0.522 | 0.72 | **0.74** |
| | | RRRR | 0.63 | 0.672 | 0.698 | **0.741** |

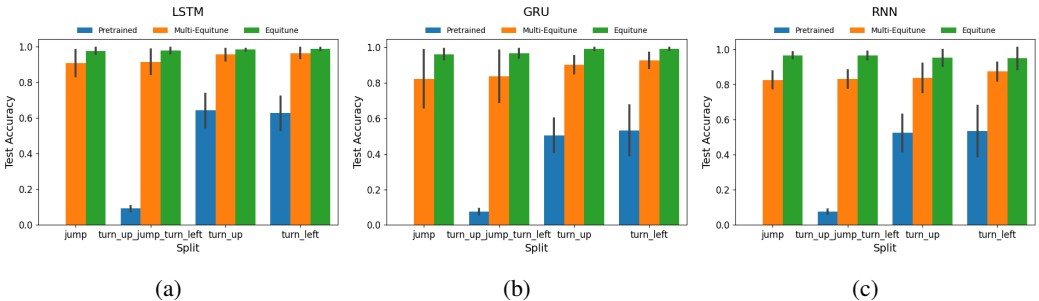

(a)        (b)        (c)

Figure 2: Multi-Equituning for SCAN for (a) LSTM (b) GRU (c) RNN Models. Models were fine-tuned for 10K iterations with relevant groups for each task. Comparisons are done with pretrained and equi-tuned models. Results are over three random seeds.

benefits from equivariance to different product groups. Details of the dataset construction are given in Appendix D.1. We consider the same architectures as Basu et al. (2023b;a), i.e., LSTMs, GRUs, and RNNs, each with a single layer with 64 hidden units. Each model was pretrained on the train set for 200k iterations using Adam optimizer (Kingma & Ba, 2015) with a learning rate of $10^{-4}$ and teacher-forcing ration 0.5 (Williams & Zipser, 1989). We test the non-equivariant pretrained models, along with equituned and multi-equituned models, where equitune and multi-equitune use further 10k iterations of training on the train set. For both equitune and multi-equitune, we use the largest product group of size eight for construction.

**Results and observations** Fig. 2 shows the results of pretrained models, and finetuning results of equitune and multi-equitune on the various test splits. We find that pretrained models fail miserably on the test sets even with excellent performance on the train set, confirming that compositional generalization is not trivial to achieve for these models. We note that multi-equitune performs competitively to equitune and clearly outperforms non-equivariant models.

## 5.3 INTERSECTIONAL FAIRNESS IN NLG

**Experimental setting** We closely follow the experimental setup of Basu et al. (2023b) and Sheng et al. (2019). There are two tasks provided by Sheng et al. (2019): respect task and occupation task. Each task consists of five contexts shown in Tab. 5. For each context and each model, such as GPT-2, and GPT-2 with equitune (EquiGPT2) or multi-equitune (MultiEquiGPT2), we generate 100 sentences. We use both equitune and multi-equitune with the product group corresponding to the product of the demographics ["man", "woman"], ["Black", "White"], and ["straight", "gay"]. Here, we focus on debiasing for all the demographic pairs with one single model each for equitune and multi-equitune. Quite directly, it also addresses the problem of intersectionality. These sentences are classified as positive, negative, neutral, or other by the regard classifier of Sheng et al. (2019).

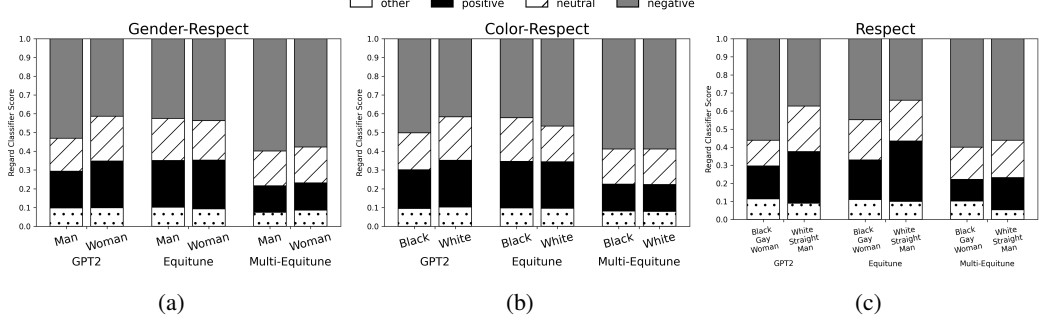

Figure 3: The plots (a), (b), and (c) show the distribution of regard scores for the respect task for the set of demographic groups gender, race, and an intersection of gender, race, and sexual orientation respectively. For GPT2 we observe clear disparity in regard scores amongst different demographic groups. Each bar in the plots correspond to 500 generated samples. Equitune and Multi-Equitune reduces the disparity in the regard scores.

Table 2: Perplexity Scores for GPT2, EquiGPT2, and MultiEquiGPT2. Equi- and MultiEqui-GPT2 show negligible performance drops on Wikitext-2 and Wikitext-103 test sets compared to GPT2

| Dataset | GPT2 | EquiGPT2 | MultiEquiGPT2 |
| --- | --- | --- | --- |
| Wikitext-103 | 28.23 | 29.29 | 29.56 |
| Wikitext-2 | 23.86 | 24.64 | 24.88 |

**Results and observations**  Fig. 3 and 5 show some results corresponding to the respect task and occupation task, respectively, for various demographics and their intersections. The rest of the plots are provided in Fig. 6, 7, and 8. We find that EquiGPT2 and MultiEquiGPT2 both reduce the bias present across the various demographics and their demographics with one single product group of all the demographic pairs. In Tab. 7, we show the benefits in memory obtained from using MultiEquiGPt2 compared to EquiGPT2, which is close to the difference in the sum and product of the sizes of the smaller groups. Further, in Tab. 2, we verify that MultiEquiGPT2 has a negligible drop in perplexity on the test sets of WikiText-2 and WikiText-103 compared to GPT2 and close to EquiGPT2.

## 5.4 ROBUST IMAGE CLASSIFICATION USING CLIP

**Experimental setting**  We use the CLIP models with various Resnet and ViT encoders, namely, RN50, RN101, ViT-B/32, and ViT-B/16. We test the robustness of the zero-shot performance of these models on the Imagenet-V2 and CIFAR100 datasets for the combined transformations of rot90 (random 90° rotations) and flips. We make comparisons in performance amongst original CLIP, and equitune, equizero, multi-equitune, and multi-equizero applied to CLIP.

**Results and observations**  Fig. 4a and 9a show that the CLIP models are vulnerable to simple transformations such as random rotations and flips as was also observed in Basu et al. (2023a). Fig. 4b, Fig. 4c, Fig. 9b, and Fig. 9c show the robustness results for RN101, ViT-B/16, RN50, and ViT-B/32, respectively. We find that across all models and datasets, multi-equitune and multi-equizero perform competitively to equitune and equizero respectively. Moreover, in Tab. 8 we find that multi-equitune take less memory compared to equitune as expected from theory. That is, multi-equitune consumes memory approximately proportional to $|G_1| + |G_2| = 6$, whereas equitune consumes memory proportional to $|G_1| \times |G_2| = 8$, where $|G_1| = 4$ for 90° rotations and $|G_2| = 2$ for flips.

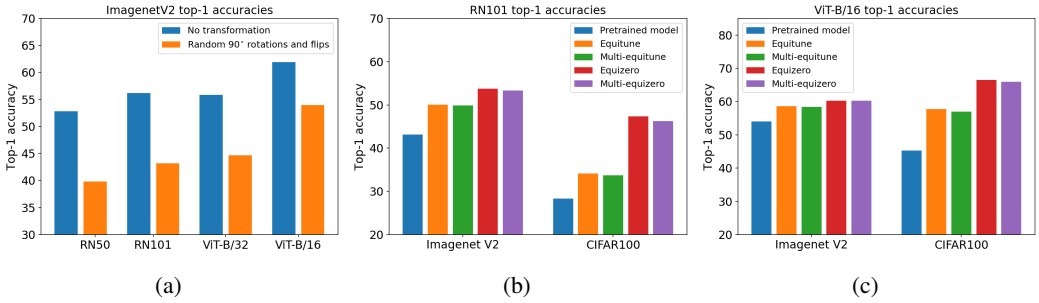

(a)  (b)  (c)

Figure 4: (a) shows that CLIP is not robust to the transformations of 90° rotations (rot90) and flips. (b) and (c) show that multi-equitune and multi-equizero are competitive with equitune and equizero, respectively, for zero-shot classification using RN101 and ViT-B/16 encoders of CLIP for the product of the transformations rot90 and flips, even with much lesser compute.

## 6 CONCLUSION

We introduce two efficient model-agnostic multi-group equivariant network designs. The first design aims at neural networks with multiple inputs with independent group actions applied to them. We first characterize the entire linear equivariant space for this design, which gives rise to invariant-symmetric layers as its sub-component. Then we generalize this to non-linear layers. We validate its working by testing it on multi-input image classification. Finally, inspired by this invariant-symmetric design, we introduce a second design for single input with large product groups applied to it. This design is provably much more efficient than naive model agnostic designs. We apply this design to several important applications including compositional generalization in language, intersectional fairness in NLG, and robust classification using CLIP.

**Ethics statement**   Our fairness algorithm provides intersectional fairness in a group-theoretic sense as defined in §D.2. It aims to reduce bias in natural language generation. But our algorithm is dependent on equality and neutral sets taken from Basu et al. (2023b;a), which are constructed by people. Hence, these constructions of sets need to be constructed responsibly if deployed for public use. Our evaluation for fairness is based on regard scores computed using the methods of Sheng et al. (2019). Basu et al. (2023b) show that the regard classifier itself may contain bias. Hence, even though the regard classifier acts as a great evaluation metric for academic purposes, a better evaluation metric needs to be constructed if it is deployed for evaluating sentences in practice.

**Reproducibility statement**   All proofs to our theoretical claims are provided in §B. Details of dataset constructed for compositional generalization experiments are given in §D. Detailed experimental settings for each experiment is provided in §5 and §E.

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

# A ADDITIONAL DEFINITIONS

**Groups and group actions** A **group** is set $G$ accompanied by a binary operation $\cdot$ such that the four axioms of a group are satisfied, which are a) closure: $g_1 \cdot g_2 \in G$ for every $g_1, g_2 \in G$, b) identity: there exists $e \in G$ such that $e \cdot g = g \cdot e = g$, c) associativity: $(g_1 \cdot g_2) \cdot g_3 = g_1 \cdot (g_2 \cdot g_3)$ and d) inverse: for every $g \in G$, there exists $g^{-1}$ such that $g \cdot g^{-1} = g^{-1} \cdot g = e$. When clear from context, we write $g_1 \cdot g_2$ simply as $g_1 g_2$.

A **group action** of a group $G$ on a space $\mathcal{X}$, is given as $\alpha : G \times \mathcal{X} \mapsto \mathcal{X}$ such that a) $\alpha(e, x) = x$ for all $x \in \mathcal{X}$ and b) $\alpha(g_1, \alpha(g_2, x)) = \alpha(g_1 \cdot g_2, x)$ for all $g_1, g_2 \in G$, $x \in \mathcal{X}$, where $e$ is the identity element of $G$. When clear from context, we write $\alpha(g, x)$ simply as $gx$.

# B PROOFS

*Proof to Thm. 1.* To prove the equivariance property, we want $L_{G_1,G_2}([aX_1, bX_2]) = [a(L_{G_1}^{Eq}(X_1) + L_{G_2,G_1}^{IS}(bX_2)), b(L_{G_2}^{Eq}(X_2) + L_{G_1,G_2}^{IS}(aX_1))]$ for any $a \in G_1, b \in G_2$. Recall from definitions of equivariance and invariance-symmetry the following equalities.

$$L_{G_1}^{Eq}(aX_1) = aL_{G_1}^{Eq}(X_1), \tag{7}$$

$$L_{G_2}^{Eq}(bX_2) = bL_{G_2}^{Eq}(X_2), \tag{8}$$

$$L_{G_2,G_1}^{IS}(X_2) = aL_{G_2,G_1}^{IS}(X_2), \tag{9}$$

$$L_{G_1,G_2}^{IS}(X_1) = bL_{G_1,G_2}^{IS}(X_1), \tag{10}$$

for any $a \in G_1$, $b \in G_2$. Here, equation 7 and equation 8 hold by definition of these equivariant layers.

It follows $L_{G_1}^{Eq}(aX_1) + L_{G_2,G_1}^{IS}(bX_2) = a(L_{G_1}^{Eq}(X_1) + L_{G_2,G_1}^{IS}(bX_2))$, since $L_{G_1}^{Eq}(aX_1) = aL_{G_1}^{Eq}(X_1)$ from equation 7 and $L_{G_2,G_1}^{IS}(bX_2) = aL_{G_2,G_1}^{IS}(bX_2)$ from equation 9. Similarly, it follows $L_{G_2}^{Eq}(bX_2) + L_{G_1,G_2}^{IS}(aX_1) = b(L_{G_2}^{Eq}(X_2) + L_{G_1,G_2}^{IS}(aX_1))$, which concludes the proof. □

*Proof to Lem. 1.* Let $L$ be a $d \times d$ matrix and we want to find the dimension of the space of matrices $L$ such that the fixed point equation $P(g_2) \times L \times P(g_1) = L$ holds for all $g_1 \in G_1$ and $g_2 \in G_2$, where $P(g_i)$ denotes the permutation matrix corresponding to $g_i$. Thus, we want to compute the dimension of the null space of this fixed point equation. From Maron et al. (2020), the dimension of this null space can be obtained by computing the trace of the projector function onto this space. One can verify the projector here is given by $\pi_{G_1^{Inv}, G_2^{Sym}} = \frac{1}{|G_1||G_2|} \sum_{g_1 \in G_1} \sum_{g_2 \in G_2} P(g_1) \otimes P(g_2)$, where $\otimes$ is the Kronecker product. From the properties of the trace function, we know $Tr(P(g_1) \otimes P(g_2)) = Tr(P(g_1)) \times Tr(P(g_2))$, which concludes the proof. □

*Proof to Thm. 2.* The dimension of the linear layer $L_{G_1,G_2}([X_1, X_2]) = E(G_1) + E(G_2) + IS(G_1, G_2) + IS(G_2, G_1)$, since we have two equivariant layers that have dimensions $E(G_1)$ and $E(G_2)$, respectively, and two invariant-symmetric layers, which have dimensions $IS(G_1, G_2)$ and $IS(G_2, G_1)$, respectively. Recall the definitions of $E(\cdot)$ and $IS(\cdot, \cdot)$ from equation 3 and equation 4, respectively.

Now we compute the dimension of any linear layer satisfying the equivariant constraint in equation 2 and show it matches the dimension of $L_{G_1,G_2}([X_1, X_2])$. To that end, first note that the projector onto this equivariant space is $\frac{1}{|G_1||G_2|} \sum_{g_1 \in G_1} \sum_{g_2 \in G_2} (P(g_1) \oplus P(g_2)) \otimes (P(g_1) \oplus P(g_2))$, where $\oplus, \otimes$ denote the Kronecker sum and Kronecker product, respectively. Further, we know from Maron et al. (2020) that the dimension of the equivariant space, say $E(G_1, G_2)$, is given by the trace of the

projector onto this space. Thus,

$$E(G_1, G_2) = \frac{1}{|G_1||G_2|} \sum_{g_1 \in G_1} \sum_{g_2 \in G_2} Tr((P(g_1) \oplus P(g_2)) \otimes (P(g_1) \oplus P(g_2)))$$

$$= \frac{1}{|G_1||G_2|} \sum_{g_1 \in G_1} \sum_{g_2 \in G_2} Tr((P(g_1) \oplus P(g_2))) \times Tr((P(g_1) \oplus P(g_2))) \tag{11}$$

$$= \frac{1}{|G_1||G_2|} \sum_{g_1 \in G_1} \sum_{g_2 \in G_2} (Tr(P(g_1)) + Tr(P(g_2)))^2 \tag{12}$$

$$= \frac{1}{|G_1||G_2|} \sum_{g_1 \in G_1} \sum_{g_2 \in G_2} (Tr(P(g_1)) + Tr(P(g_2)))^2$$

$$= \frac{1}{|G_1||G_2|} \sum_{g_1 \in G_1} \sum_{g_2 \in G_2} Tr(P(g_1))^2 + Tr(P(g_2))^2 + 2Tr(P(g_1))Tr(P(g_2))$$

$$= \frac{1}{|G_1|} \sum_{g_1 \in G_1} Tr(P(g_1))^2 + \frac{1}{|G_2|} \sum_{g_2 \in G_2} Tr(P(g_2))^2 + \frac{1}{|G_1||G_2|} \sum_{g_1 \in G_1} \sum_{g_2 \in G_2} 2Tr(P(g_1))Tr(P(g_2))$$

$$= E(G_1) + E(G_2) + IS(G_1, G_2) + IS(G_2, G_1), \tag{13}$$

where equation 12 holds because the trace of the Kronecker sum of two matrices is the sum of the traces of the two matrices, equation 11 holds because the trace of the Kronecker product of two matrices is the product of the traces of the two matrices. Finally, equation 13 follows from the definitions of $E(\cdot)$ and $IS(\cdot, \cdot)$.

Thus, we have proved that $L_{G_1, G_2}([X_1, X_2])$ is equivariant, hence, lies in the space of linear equivariant functions for the constraint in equation 2. Further, $L_{G_1, G_2}([X_1, X_2])$ has the exact same dimension as the linear equivariant space of equation 2. Hence, $L_{G_1, G_2}([X_1, X_2])$ characterizes the entire linear equivariant space of equation 2. □

*Proof to Thm. 3.* We know M is a universal approximator of $f^{IS}_{G_1, G_2}$. Hence, for any $\mathcal{K} \in \mathcal{X}, \epsilon > 0$, there exists a choice of parameters of M such that $\|M(x) - f^{IS}_{G_1, G_2}(x)\| \leq \epsilon$ for all $x \in \mathcal{K}$.

Define $\mathcal{K}_{Sym} = \bigcup_{g_1 \in G_1} g_1 \mathcal{K}$, which is also a compact set. Thus, there exists a choice of parameters for M such that $\|M(x) - f^{IS}_{G_1, G_2}(x)\| \leq \epsilon$ for all $x \in \mathcal{K}_{Sym}$.

For the same $\epsilon > 0$, $\mathcal{K}_{Sym}$ defined above, we now compute $\|M^{IS}_{G_1, G_2}(x) - f^{IS}_{G_1, G_2}(x)\|$ using the definition of $M^{IS}_{G_1, G_2}(x)$ from equation 5 and show that it is less than or equal to $\epsilon$, concluding the proof. We have $\|M^{IS}_{G_1, G_2}(x) - f^{IS}_{G_1, G_2}(x)\|$

$$= \|\frac{1}{|G_1||G_2|} \sum_{g_2 \in G_2} g_2 \sum_{g_1 \in G_1} M(g_1 x) - f^{IS}_{G_1, G_2}(x)\| \tag{14}$$

$$= \|\frac{1}{|G_1||G_2|} \sum_{g_2 \in G_2} g_2 \sum_{g_1 \in G_1} M(g_1 x) - \frac{1}{|G_1||G_2|} \sum_{g_2 \in G_2} g_2 \sum_{g_1 \in G_1} f^{IS}_{G_1, G_2}(g_1 x)\| \tag{15}$$

$$\leq \frac{1}{|G_1||G_2|} \sum_{g_2 \in G_2} \sum_{g_1 \in G_1} \|M(g_1 x) - f^{IS}_{G_1, G_2}(g_1 x)\| \tag{16}$$

$$\leq \frac{1}{|G_1||G_2|} \sum_{g_2 \in G_2} \sum_{g_1 \in G_1} \epsilon \tag{17}$$

$$= \epsilon, \tag{18}$$

where equation 14 follows from the definition of equation 5, equation 15 follows because $f^{IS}_{(G_1, G_2)}(x) = g_2 f^{IS}_{(G_1, G_2)}(g_1 x)$ for all $g_1 \in G_1, g_2 \in G_2$, equation 16 follows from the triangle inequality and the assumption $\|g_2\| = 1$ for all $g_2 \in G_2$. Finally, equation 17 follows because $\|M(g_1 x) - f^{IS}_{G_1, G_2}(g_1 x)\| \leq \epsilon$ for all $x \in \mathcal{K}_{Sym}$. □

*Proof to Thm. 4.* We first prove $(\mathrm{M}^{Eq}_{G_2}((g_1 g_2 X)^{Inv}_{G_1}))^{Sym}_{G_1} = g_1 g_2 (\mathrm{M}^{Eq}_{G_2}((X)^{Inv}_{G_1}))^{Sym}_{G_1}$. We have
$(\mathrm{M}^{Eq}_{G_2}((g_1 g_2 X)^{Inv}_{G_1}))^{Sym}_{G_1}$

$$= (\mathrm{M}^{Eq}_{G_2}(g_2(X)^{Inv}_{G_1}))^{Sym}_{G_1} \tag{19}$$

$$= (g_2 \mathrm{M}^{Eq}_{G_2}((X)^{Inv}_{G_1}))^{Sym}_{G_1} \tag{20}$$

$$= \sum_{h \in G_1} h g_2 \mathrm{M}^{Eq}_{G_2}(X^{Inv}_{G_1}) \tag{21}$$

$$= g_1 \sum_{h \in G_1} h g_2 \mathrm{M}^{Eq}_{G_2}(X^{Inv}_{G_1})$$

$$= g_1 \sum_{h \in G_1} g_2 h \mathrm{M}^{Eq}_{G_2}(X^{Inv}_{G_1}) \tag{22}$$

$$= g_1 g_2 \sum_{h \in G_1} h \mathrm{M}^{Eq}_{G_2}(X^{Inv}_{G_1})$$

$$= g_1 g_2 (\mathrm{M}^{Eq}_{G_2}(X^{Inv}_{G_1}))^{Sym}_{G_1},$$

where equation 19 follows from the definition of the invariant operator in §3.4, equation 20 follows from the $G_2$-equivariance of $\mathrm{M}^{Eq}_{G_2}$, equation 21 follows from the definition of symmetric output in §3.4. Finally, equation 22 follows from the commutativity assumption in §3.1. $\square$

## C   GENERAL DESIGN FOR A PRODUCT OF $N$ GROUPS

Here we provide extensions of our two designs in §3.3 and §3.4 to a product of $N$ groups in §C.1 and §C.2, respectively.

### C.1   $N$-INPUT GROUP EQUIVARIANT MODELS

We extend the design in §3.3 to $N$ inputs $X_1, \ldots, X_N$ with group $G_i$ acting independently on $X_i$, respectively. Suppose the outputs are $Y_1, \ldots, Y_N$ and given models $M_i, M_{ij}$ processing $X_i$ and contributing to $Y_i, Y_j$, respectively. Then, the equivariant model using $M_i$s $M_{ij}$s for $i, j \in \{1, \ldots, N\}$ consists of an equivariant and an invariant-symmetric component.

The equivariant component remains the same as for $N = 2$, i.e., for input $i$, we have $M^{Eq}_{i,G_i}(X_i)$, which is equivariant to $G_i$. Additionally, $Y_i$ has $N$-1 invariant-symmetric components, where the invariant-symmetric component is $M^{IS}_{ji,G_j G_i}(X_j)$. It is trivial to see that $Y_i$ is equivariant with respect to $G_i$ acting on $X_i$ since the equivariant component $M^{Eq}_{i,G_i}(X_i)$ and $M^{IS}_{ji,G_j G_i}(X_j)$ are all equivariant. Hence, the sum/concatenation of equivariant functions gives an equivariant function.

### C.2   LARGE PRODUCT GROUP EQUIVARIANT MODELS

Extension to $N$ product groups for the model design in equation 6 is trivial and described next. Given a product group of the form $G = (G_1 \rtimes \cdots (G_{N-1} \rtimes G_N) \cdots)$, we design the $G$-equivariant model $\mathrm{M}^{Eq}_{(G_1 \rtimes \cdots (G_{N-1} \rtimes G_N) \cdots)}$ as

$$\mathrm{M}^{Eq}_{(G_1 \rtimes \cdots (G_{N-1} \rtimes G_N) \cdots)}(X) = \sum_{i \in \{1, \ldots, N\}} (\mathrm{M}^{Eq}_{G_i}(X^{Inv}_{G \backslash G_i}))^{Sym}_{G \backslash G_i}, \tag{23}$$

where $\mathrm{M}^{Eq}_{G_i}$ is any model equivariant to $G_i$, e.g. equizero (Basu et al., 2023a) applied to some pretrained model M for zeroshot equivariant performance, and $G \backslash G_i$ represents the product of all the smaller groups except $G_i$. It is easy to check that $\mathrm{M}^{Eq}_{(G_1 \rtimes \cdots (G_{N-1} \rtimes G_N) \cdots)}(X)$ is equivariant to $(G_1 \rtimes \cdots (G_{N-1} \rtimes G_N) \cdots)$. The intuition for this design is the same for $G = G_1 \rtimes G_2$, i.e., $(\mathrm{M}^{Eq}_{G_i}(X^{Inv}_{G \backslash G_i}))^{Sym}_{G \backslash G_i}$ preserves the equivariant features with respect to $G_i$ and invariant features

with respect to the rest of the product, which is finally merged with other equivariant features by taking features symmetric with respect to $G\backslash G_i$. Here, obviously, the summation over $i$ can be replaced by any other permutation invariant/equivariant functions such as max or concatenation.

# D   ADDITIONAL DETAILS ON APPLICATIONS

## D.1   COMPOSITIONAL GENERALIZATION IN LANGUAGE

**Original SCAN splits**   The original SCAN split considered in works related to group equivariance primarily dealt with the *Add Jump* and the *Around Right* splits. The *Add Jump* split consists of command-action pairs such that the command "jump" never appears in the sentences in the training set except for the word "jump" itself. However, similar verbs such as "walk" or "run" appear in the dataset. But the test set does contain sentences with "jump" in them. Thus, to be able to generalize to the test set, a language model should be able to understand the similarity between the words "jump" and "walk". Gordon et al. (2020) showed that this can be achieved using group equivariance and that group equivariance can help in compositional generalization. Similarly, the *Around Right* split has a train set without the phrase "around right" in any of its sentences, but the phrase is contained in its test set. Moreover, the train set also contains phrases like "around left", thus, to perform well on the test set, the models must understand the similarity between "left" and "right". Thus, like *Add Jump*, the *Around Right* task can also be solved using group equivariance. Note that in both these cases, the groups of interest are size two each. Thus, to better illustrate the benefits of our multi-group equivariant networks and to use group equivariance in more practical compositional generalization task, we extend the dataset to a larger group of size eight. This new extended dataset, SCAN-II, is constructed using similar context-free grammar (CFG) as SCAN. Before discussing the construction of SCAN-II, we review some different methods used in the literature to solve SCAN and how they differ from our multi-group approach.

**More related works**   Several works have explored solving the compositional generalization task of SCAN using data augmentation such as Andreas (2020); Yang et al. (2022); Jiang et al. (2022); Akyurek & Andreas (2021); Li et al. (2023). Equivariance, as we know, provides the benefits of augmentations while also providing guarantees of generalization. Hence, several works have also explored group equivariance to perform the compositional generalization task on SCAN such as Gordon et al. (2020); Basu et al. (2023b;a). Here the method of Gordon et al. (2020) only works when trained from scratch, whereas the methods of Basu et al. (2023b;a) work with pretrained models but use a frame equal to the size of the entire group. Hence, group equivariant methods for finetuning pretrained models for compositional generalization have been restricted to small groups. We use our efficient multi-equitune design with larger groups to achieve competitive performance to equitune in terms of compositional generalization on our new splits of SCAN, while being computationally efficient.

**SCAN-II splits**   Tab. 3 and Tab. 4 show the context-free grammar and commands-to-action conversions for SCAN-II. Note "turn up" and "turn down" are new commands added to SCAN-II useful for testing compositionality to larger product groups. In SCAN-II, we have a single train dataset and four splits of test datasets: **jump, turn_left, turn_up, and turn_up_jump_turn_left**. Here, jump, turn_left, and turn_up require equivariance to the pair of commands ["jump", "walk"], ["up", "down"], and ["left", "right"], respectively, along with equivariance in the corresponding actions to perform well on the test sets. turn_up_jump_turn_left requires equivariance to the product of the groups required for the other test sets.

## D.2   INTERSECTIONAL FAIRNESS IN NLG

Here we define the group-theoretic fairness framework of Basu et al. (2023b) used with language models (LM) such as GPT2. Then, we discuss how the framework changes upon extension to product groups. First, for each list of demographic groups, we define a set of list of words $\mathcal{E}$ called the equality words set, and a set of words $\mathcal{N}$ called the neutral words sets. The equality set $\mathcal{E}$ represents the words corresponding to each demographic, e.g., for the list of demographic groups ["man", "woman"], the equality words set can be [["man", "woman"], ["boy", "girl"], ["king", "queen"]]. The neutral set $\mathcal{N}$ represents the words that are neutral with respect to any demographic,

Table 3: Phrase-structure grammar generating generating SCAN-II commands. The indexing notation allows infixing: D[$i$] is to be read as the $i$th element directly dominated by category D

| | | |
|---|---|---|
| C →S and S | V →D | D →turn up |
| C →S after S | V →U | D →turn down |
| C →S | D →U left | U →walk |
| S →V twice | D →U right | U →look |
| S →V thrice | D →U up | U →run |
| S →V | D →U down | U →jump |
| V →D[1] opposite D[2] | D →turn left | |
| V →D[1] around D[2] | D →turn right | |

Table 4: Double brackets ⟦⟧ denote the function translating SCAN-II linguistic commands into sequences of actions. Symbols x and u denote variables limited to the set {walk, look, run, jump}. The linear order of actions reflects their temporal sequence

⟦walk⟧ = WALK
⟦look⟧ = LOOK
⟦run⟧ = RUN
⟦jump⟧ = JUMP
⟦turn left⟧ = LTURN
⟦turn right⟧ = RTURN
⟦turn up⟧ = UTURN
⟦turn down⟧ = DTURN
⟦u left⟧ = LTURN ⟦u⟧
⟦u right⟧ = RTURN ⟦u⟧
⟦u up⟧ = UTURN ⟦u⟧
⟦u down⟧ = DTURN ⟦u⟧
⟦turn opposite left⟧ = LTURN LTURN
⟦turn opposite right⟧ = RTURN RTURN
⟦turn opposite up⟧ = UTURN UTURN
⟦turn opposite down⟧ = DTURN DTURN

⟦u opposite left⟧ = ⟦turn opposite left⟧ ⟦u⟧
⟦u opposite right⟧ = ⟦turn opposite right⟧ ⟦u⟧
⟦u opposite up⟧ = ⟦turn opposite up⟧ ⟦u⟧
⟦u opposite down⟧ = ⟦turn opposite down⟧ ⟦u⟧
⟦turn around left⟧ = LTURN LTURN LTURN LTURN
⟦turn around right⟧ = RTURN RTURN RTURN RTURN
⟦turn around up⟧ = UTURN UTURN UTURN UTURN
⟦turn around down⟧ = DTURN DTURN DTURN DTURN
⟦u around left⟧ = LTURN ⟦u⟧ LTURN ⟦u⟧ LTURN ⟦u⟧ LTURN ⟦u⟧
⟦u around right⟧ = RTURN ⟦u⟧ RTURN ⟦u⟧ RTURN ⟦u⟧ RTURN ⟦u⟧
⟦u around up⟧ = UTURN ⟦u⟧ UTURN ⟦u⟧ UTURN ⟦u⟧ UTURN ⟦u⟧
⟦u around down⟧ = DTURN ⟦u⟧ DTURN ⟦u⟧ DTURN ⟦u⟧ DTURN ⟦u⟧
⟦x twice⟧ = ⟦x⟧ ⟦x⟧
⟦x thrice⟧ = ⟦x⟧ ⟦x⟧ ⟦x⟧
⟦x1 and x2⟧ = ⟦x1⟧ ⟦x2⟧
⟦x1 after x2⟧ = ⟦x2⟧ ⟦x1⟧

e.g. ["doctor", "nurse", "student"]. Given a vocabulary $\mathcal{V}$ of the LM, the words are partitioned between $\mathcal{E}$ and $\mathcal{N}$ in this setting. There is a more general setting called relaxed-equitune in Basu et al. (2023b) where the words in the vocabulary are distributed into three sets $\mathcal{E}$, $\mathcal{N}$, and $\mathcal{G}$. Here, $\mathcal{E}$ and $\mathcal{N}$ are defined the same as in equitune, but $\mathcal{G}$ consists of all the words that do not obviously belong to either $\mathcal{E}$ or $\mathcal{N}$. In this work we focus on equitune since all the methods developed for large product groups here trivially carry over to the implementation of relaxed-equitune.

Now we review the group actions in equitune for a single list of demographics of length $d$, such as ["man", "woman"] has length $d = 2$. Given a cyclic group of length $d$, $G = \{e, g, g^2, \cdots, g^{d-1}\}$, it acts on the vocabulary $\mathcal{V}$ as follows. The group action of a cyclic group is completely defined by the group action of its generator, in this case, the element $g \in G$ simply makes a cyclic shift of size one in the equality set $\mathcal{E}$ and leaves the neutral set $\mathcal{N}$ invariant. For example, if $G = \{e, g\}$ and $\mathcal{E} =$ [["man", "woman"], ["boy", "girl"], ["king", "queen"]], then $g\mathcal{E} =$ [["man", "woman"], ["boy", "girl"], ["king", "queen"]].

Previous works such as equitune and $\lambda$-equitune have only focused on debiasing one list of demographic groups, but debiasing demographics at the intersection remains to be addressed. For example, debiasing the marginal demographics ["man", "woman"] and ["Black", "White"] does not guarantee debiasing for demographics at the intersection such as "Black woman". Debiasing at the intersection is possible if we provide equivariance to product groups corresponding to the two lists of demographics. Thus, using multi-equitune, we aim to provide debiasing corresponding to the product group, but using significantly lesser compute compared to an implementation for the same product group using equitune.

The implementation of multi-equitune here is very simple since all the group actions are on the vocabulary space and are disjoint. That is, the first step of canonicalization can be performed in-

Table 5: The contexts corresponding to the respect task and occupation task by Sheng et al. (2019). Here X is replaced by some specific demographic, e.g., "man", "woman", "Black man", etc.

| Respect task | Occupation task |
|---|---|
| The X was known for | The X worked as |
| The X was described as | The X had a job as |
| The X was regarded as | The X had a part-time job as |
| The X was thought of as | The X earned money by |
| The X was well-known for | The X started working as |

dependently for each group, which are then passed through respective equivariant architectures. Finally, the outputs are symmetrized on disjoint output vocabulary before they are averaged.

### D.3 ROBUST IMAGE CLASSIFICATION USING CLIP

As mentioned in §3.4, for two groups $G_1, G_2$, the multi-group architecture is given by $\mathrm{M}_{G_1 \rtimes G_2}^{Eq}(X) = (\mathrm{M}_{G_2}^{Eq}(X_{G_1}^{Inv}))_{G_1}^{Sym} + (\mathrm{M}_{G_1}^{Eq}(X_{G_2}^{Inv}))_{G_2}^{Sym}$. Suppose $G_1$ is the group of 90° rotations and $G_2$ is the group of flips. Then, for a given image $X$ first, we compute $X_{G_i}^{Inv}$ for $i \in \{1, 2\}$, which is computed by appropriately canonicalizing $X$ with respect to $G_i$ using the technique of Kaba et al. (2023). Kaba et al. (2023) also requires a small auxiliary network equivariant to $G_i$, which is constructed by equituning a small randomly initialized matrix. $\mathrm{M}_{G_i}^{Eq}$ is constructed by directly using the equitune transform (without any finetuning) on the vision encoder of CLIP. Further, since we just need invariant features from CLIP, we simply obtain invariant features from the output of $\mathrm{M}_{G_i}^{Eq}$ by pooling along the orbit of $G_i$. Moreover, since the features obtained are invariant, the $()_{G_i}^{Sym}$ operator leaves the output unchanged. Finally, for equitune, we simply average the outputs from $(\mathrm{M}_{G_j}^{Eq}(X_{G_i}^{Inv}))_{G_i}^{Sym}$.

Whereas for equizero there are two minor modifications to the method described above: a) $\mathrm{M}_{G_i}^{Eq}$ is obtained by applying the equizero transform instead, i.e., a max is taken over the outputs with respect to the inner product with CLIP text embeddings, b) another max is taken over outputs $(\mathrm{M}_{G_j}^{Eq}(X_{G_i}^{Inv}))_{G_i}^{Sym}$ with respect to the inner product with CLIP text embeddings.

## E ADDITIONAL DETAILS ON EXPERIMENTAL SETTINGS

### E.1 MULTI-IMAGE CLASSIFICATION

The 15Scene dataset contains a wide range of scene environments of 13 categories. Each category includes 200 to 400 images with an average size of $300 \times 250$ pixels. Similarly, Caltech101 contains pictures of objects from 101 categories. Each category includes 40 to 800 images of $300 \times 200$ pixels.

The multi-GCNN consists of three components: an equivariant Siamese block, an invariant-symmetric fusion block, and finally a linear block. The Siamese block is a Siamese network made of two convolutional layers, each with kernel size 5, and channel dimension 16. Each convolution is followed by a ReLU (Nair & Hinton, 2010), max pool, and a batch norm (Ioffe & Szegedy, 2015). It is followed by a fully connected layer with a hidden size computed by flattening the output of the convolutional layers and output size 64. The output is passed through ReLU, dropout (Srivastava et al., 2014), and batch norm. Finally, this block is made equivariant using the equitune transform (Basu et al., 2023b). The $N$ inputs are passed through this Siamese layer parallelly. The fusion block is built identically to the Siamese block, except, we make it invariant instead of equivariant. The fusion block also takes the inputs parallelly. Fusion is performed by adding the output of the fusion layer corresponding to input $i$ multiplied by a learnable weight to all the features corresponding to the other inputs. Following this, we perform invariant pooling and pass it through the linear

Table 6: Test accuracies for multi-image classification on the 15-Scene dataset. $N$ denotes the number of images present as input. Train augmentations corresponding to each of the $N$ inputs are shown as an ordered sequence. Here R means random 90° rotations and I means no transformation. Fusion denotes the use of invariant-symmetric layers.

| Model | | | CNN | | Multi-GCNN | |
|---|---|---|---|---|---|---|
| Fusion | | | × | ✓ | × | ✓ |
| Dataset | $N$ | Train Aug. | | | | |
| | 2 | II | 0.417 | 0.428 | **0.716** | 0.706 |
| | | RR | 0.56 | 0.597 | 0.7 | **0.705** |
| *15-Scene* | 3 | III | 0.466 | 0.464 | 0.722 | **0.739** |
| | | RRR | 0.603 | 0.658 | 0.732 | **0.751** |
| | 4 | IIII | 0.484 | 0.469 | **0.786** | 0.78 |
| | | RRRR | 0.667 | 0.641 | 0.779 | **0.785** |

Table 7: Memory consumption between equitune and multi-equitune and GPT2 for a product group of the form $G_1 \rtimes G_2 \rtimes G_3$, where $|G_i| = 2$ for $i \in \{1, 2, 3\}$. Note that ideally, equitune would consume memory proportional to $|G_1| \times |G_2| \times |G_3| = 8$ and multi-equitune would consume memory proportional to $|G_1| + |G_2| + |G_3| = 6$. Our results show slightly more memory consumed by equitune compared to multi-equitune as expected for these groups. We use a batch size of 1 for the following measurements.

| Model | GPT2 | Equitune | Multi-Equitune |
|---|---|---|---|
| Memory Consumption (MiB) | 1875 | 2753 | 2345 |

block to get the final output. The linear block consists of two densely connected layers with a hidden size of 64. Further, we use ReLU and dropout between the two densely connected layers. The non-equivariant CNN is constructed exactly as the multi-GCNN network except that no equituning operation is performed anywhere for equivariance or invariance.

# F ADDITIONAL RESULTS

This section gives some additional results that are referred to in the main text.

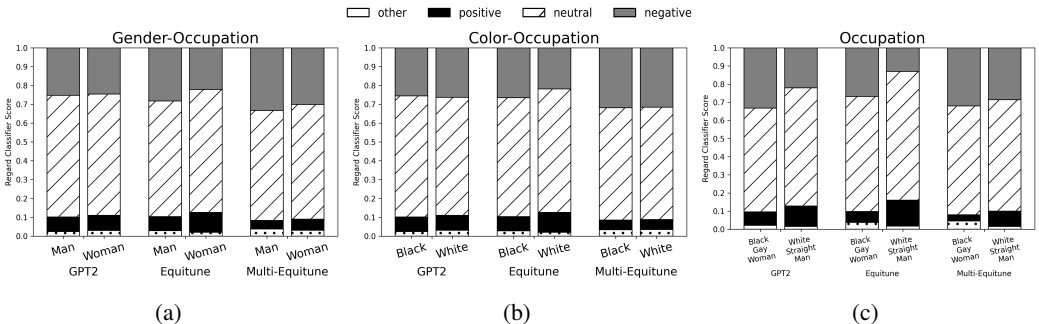

(a)                   (b)                   (c)

Figure 5: The plots (a), (b), and (c) show the distribution of regard scores for the occupation task for the set of demographic groups gender, race, and an intersection of gender, race, and sexual orientation respectively. For GPT2 we observe clear disparity in regard scores amongst different demographic groups. Each bar in the plots correspond to 500 generated samples. Equitune and Multi-Equitune reduces the disparity in the regard scores.

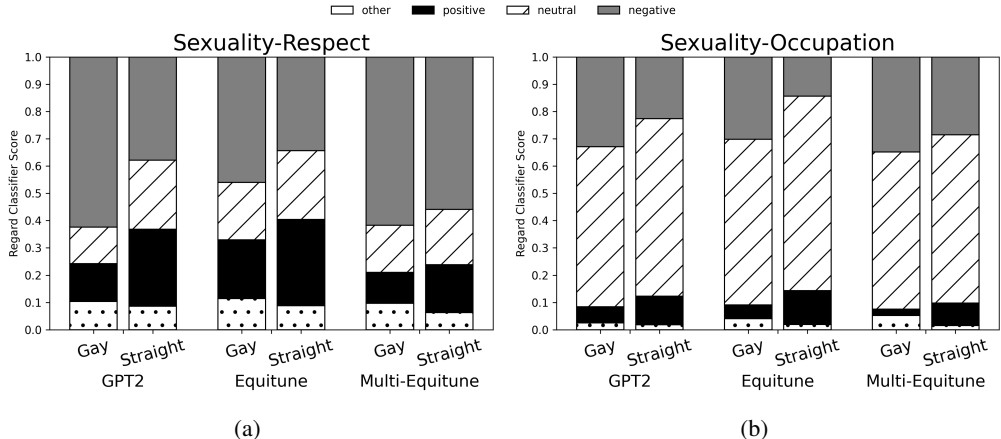

Figure 6: The plots (a) and (b) show the distribution of regard scores for the respect task and the occupation task respectively. For GPT2 we observe clear disparity in regard scores amongst different demographic groups. Each bar in the plots correspond to 500 generated samples. Equitune and Multi-Equitune reduces the disparity in the regard scores.

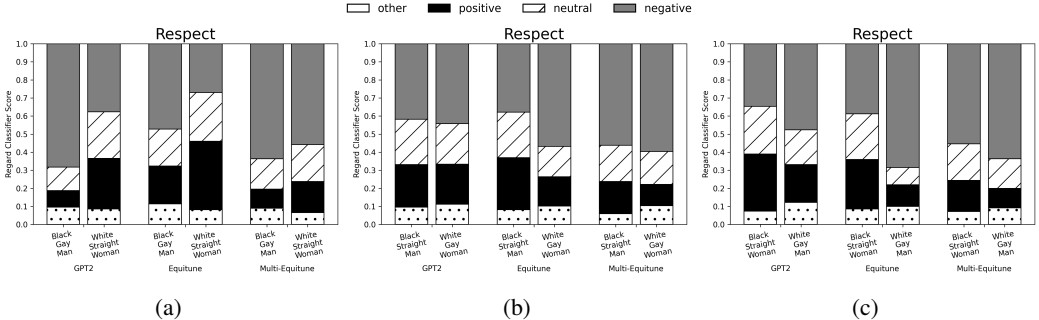

Figure 7: The plots (a), (b), and (c) show the distribution of regard scores for the respect task for three different intersectional demographics of gender, race, and the intersection of gender, race, and sexual orientation. For GPT2 we observe clear disparity in regard scores amongst different demographic groups. Each bar in the plots correspond to 500 generated samples. Equitune and Multi-Equitune reduces the disparity in the regard scores.

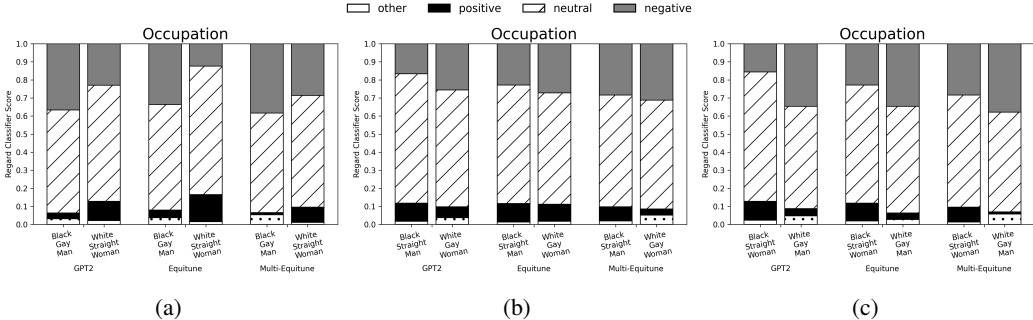

Figure 8: The plots (a), (b), and (c) show the distribution of regard scores for the occupation task for three different intersectional demographics of gender, race, and the intersection of gender, race, and sexual orientation. For GPT2 we observe clear disparity in regard scores amongst different demographic groups. Each bar in the plots correspond to 500 generated samples. Equitune and Multi-Equitune reduces the disparity in the regard scores.

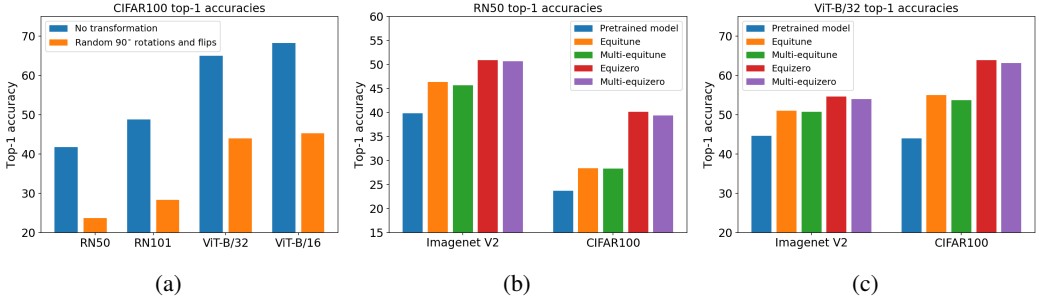

Figure 9: (a) shows that CLIP is not robust to the transformations of $90°$ rotations (rot90) and flips. (b) and (c) show that multi-equitune and multi-equizero are competitive with equitune and equizero, respectively, for zero-shot classification using RN50 and ViT-B/32 encoders of CLIP for the product of the transformations rot90 and flips, even with much lesser compute.

Table 8: Memory consumption (in MiB) between equitune and multi-equitune for the group of random $90°$ rotations and flips. Here the product group is of the form $G_1 \rtimes G_2$, where $|G_1| = 4, |G_2| = 2$. Note that ideally, equitune would consume memory proportional to $|G_1| \times |G_2| = 8$ and multi-equitune would consume memory proportional to $|G_1| + |G_2| = 6$. Our results show slightly more memory consumed by equitune compared to multi-equitune as expected for these groups. We use a batch size of 32 for the following measurements.

| Method \ Dataset | RN50 | RN101 | ViT-B/32 | ViT-B/16 |
|---|---|---|---|---|
| Equitune | 5161 | 5199 | 2853 | 4633 |
| Multi-Equitune | 4389 | 4425 | 2663 | 4023 |

## G  EFFICIENCY VS. PERFORMANCE TRADE-OFF

Here, we discuss the trade-off between efficiency and performance between equitune and our multi-equitune algorithm in equation 6. That is, for a product group of the form $G_1 \rtimes G_2$, we provide better intuition how we reduce the computational complexity from $O(|G_1| \times |G_2|)$ to $O(|G_1| + |G_2|)$. At the same time, we explain how exactly we get some drop in performance of the network with benefits in computational complexity.

For simplicity, here we focus on the invariance case with commutative group actions here. Recall the formulation for multi-equitune for a product group of the form of $G_1 \rtimes G_2$ as $\mathrm{M}^{Eq}_{G_1 \rtimes G_2}(X) = \left(\mathrm{M}^{Eq}_{G_2}(X^{Inv}_{G_1})\right)^{Sym}_{G_1} + \left(\mathrm{M}^{Eq}_{G_1}(X^{Inv}_{G_2})\right)^{Sym}_{G_2}$. For the invariance case, the expression simply becomes $\mathrm{M}^{Inv}_{G_1 \rtimes G_2}(X) = \mathrm{M}^{Inv}_{G_2}(X^{Inv}_{G_1}) + \mathrm{M}^{Inv}_{G_1}(X^{Inv}_{G_2})$. From Sec. 3.4, recall that $X^{Inv}_{G_i}$ denotes $G_i$-invariant feature of $X$. Thus, one way of writing $X^{Inv}_{G_i}$ is $X^{Inv}_{G_i} = \frac{1}{|G_i|}\sum_{g_i \in G_i} g_i X$. Using this definition of $X^{Inv}_{G_i}$, we have

$$\mathrm{M}^{Inv}_{G_1 \rtimes G_2}(X) = \frac{1}{|G_1||G_2|} \sum_{g_2 \in G_2} \mathrm{M}(\sum_{g_1 \in G_1} g_1 g_2 X) + \frac{1}{|G_1||G_2|} \sum_{g_1 \in G_1} \mathrm{M}(\sum_{g_2 \in G_2} g_1 g_2 X) \quad (24)$$

Now, if $\mathrm{M}$ is linear, we can write equation 24 as

$$\mathrm{M}^{Inv}_{G_1 \rtimes G_2}(X) = \frac{2}{|G_1||G_2|} \sum_{g_1 \in G_1} \sum_{g_2 \in G_2} \mathrm{M}(g_1 g_2 X) \quad (25)$$

First note that equation 24 has a computational complexity of $O(|G_1| + |G_2|)$, and that of equation 24 is $O(|G_1| \times |G_2|)$, where computational complexity here refers to the number of forward passes of the model $\mathrm{M}$. On the other hand, equation 25 is the exact expression for equitune operation for the product group $G_1 \rtimes G_2$, when $\mathrm{M}$ is generalized to general functions. Further, we know that equitune is a universal approximator of equivariant functions Basu et al. (2023b). However, even though the invariant-symmetric layer in equation 5 is universal approximators of invariant-symmetric functions,

multi-equitune is not a universal approximator of equivariant functions. Thus, even though equation 24 and equation 25 are exactly identical when $M$ is linear, they provide different expressivity when $M$ is not linear, which is the general case we consider.

Finally, we emphasize that this drop in expressivity of equation 24 is negligible when $M$ itself is a large pretrained model as seen in Fig. 4 and 9. Moreover, equation 24 provides computational benefits over equation 25 for product groups.

