# OpenReview forum: "Efficient Model-Agnostic Multi-Group Equivariant Networks"
_ICLR.cc/2024/Conference — Submitted to ICLR 2024_

### Official Review · Reviewer_mvp1 · 2023-10-30

**Soundness:** 2 fair
**Presentation:** 2 fair
**Contribution:** 2 fair
**Rating:** 5
**Confidence:** 3

**Summary:**

The paper considers the problem of designing model-agnostic group-equivariant networks in an efficient manner. The authors consider two settings: (1) the network has multiple inputs each with a potentially different group acting on it, and (2) the network has a single input and the group acting on it is a large product group. For the former the authors consider linear formulations and characterize the entire space of linear equivariant layers. They then use the obtained equations to extend to non-linear models and show that there exist a design that is universal in approximating invariant-symmetric functions. For the second setting, they propose a method that is more efficient than existing works, at the cost of decreased expressivity.

**Strengths:**

I think the motivation is strong and the theoretical results are valuable.

**Weaknesses:**

I think the group-equivariant network that is designed for inputs having large product groups acting on them is claimed to be less expressive but more efficient than equitune (page 2). However the loss of expressivity is not discussed in section 3. I think this part is fundamental and should be stressed.

**Questions:**

1. For the results in Table 1, why do you consider the input as ordered? What if instead we consider the input as a set of images, and apply Maron et al 2020? How would it compare?
2. Related to 1, I think the multi-image task does not really test the first scenario, as the input groups are the same and therefore Maron et al 2020 can be directly applied.

---

> ### Author Response · Authors · 2023-11-22
> **Response to reviewer mvp1**
>
> We thank the reviewer for their feedback. We aim to address the weaknesses here.
>
> **Reviewer:** Weaknesses:
> I think the group-equivariant network that is designed for inputs having large product groups acting on them is claimed to be less expressive but more efficient than equitune (page 2). However the loss of expressivity is not discussed in section 3. I think this part is fundamental and should be stressed.
>
> **Response:** Thanks for this comment.  We have now added a discussion on the loss of expressivity in the newly added Sec. G in the appendix of the paper to provide an intuition describing where the trade-off between efficiency and performance comes from.
>
> **Reviewer:**
> Questions:
> For the results in Table 1, why do you consider the input as ordered? What if instead we consider the input as a set of images, and apply Maron et al 2020? How would it compare?
>
> **Response:** If we ensure that the fusion layers are identical and share across channels, then our framework directly extends for sets instead of sequences. But Maron et al. 2020 is not applicable here since they assume that the group elements acting on each image is the same. Thus, Maron et al. is not equivariant to the group actions considered here. Whereas, in Tab. 1, the group elements applied are chosen independently.
>
> In the appendix of Maron et al. Sec. B, they also provide some results on applying independent group actions to each image/data point, but they assume the group actions are transitive. We make no such assumptions. In fact, the transitive assumption does not hold on the image classification tasks considered.
>
> **Reviewer:** Related to 1, I think the multi-image task does not really test the first scenario, as the input groups are the same and therefore Maron et al 2020 can be directly applied.
>
> **Response:** As discussed above, please note the framework of Maron et al. 2020 does not work in our problem setting.

---

### Official Review · Reviewer_EpCD · 2023-11-01

**Soundness:** 3 good
**Presentation:** 2 fair
**Contribution:** 2 fair
**Rating:** 5
**Confidence:** 2

**Summary:**

This paper tries to address the computational problem of constructing model-agnostic group equivariant networks for large product groups, and provides efficient model-agnostic equivariant designs for two related problems with different input specifications. For different problems, this paper proposes new fusion layer designs, which can be extended beyond linear models, and model-agnostic equivariant designs for large product groups. Experimental results are provided for different applications, such as language compositionality, natural language generation, and zero-shot classification, showing high computational efficiency than the existing ones.

**Strengths:**

This paper is well-organized and well-written. The motivation is stated in a clear way, and the objective is easy to follow.

The theoretical findings are organized in a proper way, and the proofs are given in a rigorous way.

The computational complexity could have been given in detail.

**Weaknesses:**

The related work could be expanded to highlight the difference between this work and the existing ones. The novelty and contributions could have been highlighted.

More institutions and explanations of the theoretical findings could be provided after each theorem. It is not easy to figure out how important these theoretical results are and how they could be used to guide the practical designs.

In addition to the comparison of computational complexity, it is unclear how and to what extent the proposed designs are practically useful.

**Questions:**

Add more intuitions and explanations for theoretical findings and possible insights to guide practical designs.

Add more meaningful experimental results to show the practical usefulness of the proposed designs.

---

> ### Author Response · Authors · 2023-11-22
> **Response to reviewer EpCD**
>
> We thank the reviewer for their feedback. We aim to address the weaknesses here.
>
> **Reviewer:** The related work could be expanded to highlight the difference between this work and the existing ones. The novelty and contributions could have been highlighted.
>
> **Response:** Thanks for the suggestion on emphasizing novelty and contributions, which makes it easier for the reader.  As we have now added in Sec. 2, our work follows the very recent trend of works in equivariant deep learning that focuses on making pretrained models equivariant [1-4].
>
> Since it a new area of research, there are very few works in this direction. Most of the works described in Sec. 2 focus on training from scratch and are therefore very different from our setup.
>
> Compared to [1] that uses a simple averaging over the entire group to obtain symmetrization, we simply perform averaging over subgroups when the group can be decomposed as products. [2-4] use weighted averaging over group elements to obtain symmetrization, which are complementary to our work and can be used on top of our work for future work.
>
> [1] Basu et al. "Equi-tuning: Group equivariant fine-tuning of pretrained models." AAAI 2023.
>
> [2] Basu et al. "Equivariant few-shot learning from pretrained models." NeurIPS 2023
>
> [3] Kim et al. "Learning probabilistic symmetrization for architecture agnostic equivariance." NeurIPS, 2023.
>
> [4] Mondal et al. "Equivariant Adaptation of Large Pre-Trained Models." NeurIPS 2023.
>
> **Reviewer:** More institutions and explanations of the theoretical findings could be provided after each theorem. It is not easy to figure out how important these theoretical results are and how they could be used to guide the practical designs.
> In addition to the comparison of computational complexity, it is unclear how and to what extent the proposed designs are practically useful.
>
> **Response:** We have now added further description in Sec. 3.2 about the connections between Thm. 1 and Thm. 2. Thm. 1 proves the equivariance of the construction in (1) and Thm. 2 shows that the construction covers the entire linear equivariant space. Hence, (1) characterizes the entire space of linear equivariant networks.
>
> Thm. 3 proves the universality of the IS fusion layer and Thm. 4 proves the equivariance of the general non-linear construction in (6), as described in Sec. 3.
>
> The main goal of the work is to provide computational efficiency over [1] with little loss in performance, as described above. Further details on the trade-off between efficiency and performance is described in the newly added Sec. G in the appendix.

---

### Official Review · Reviewer_gLGd · 2023-11-03

**Soundness:** 1 poor
**Presentation:** 2 fair
**Contribution:** 1 poor
**Rating:** 3
**Confidence:** 2

**Summary:**

The paper aims at designing a model-agnostic group equivariant network for direct product groups.

**Strengths:**

The topic is interesting and important.

**Weaknesses:**

It could be that I misunderstand something. But from what I understand from the construction, the model is vacuous, and can only apply for the trivial symmetry.

In page 2, the notion of “invariant-symmetric” seems strange. Note that you simply define the identity action of $G_2$ on the range of $f$ in the space $Y$. Namely, for any $y=f(x)\in Y$, you assume that $g_2y=y$. In what sense is this an interesting model of symmetries? It describes no symmetry, or more accurately, just the trivial symmetry.

In the construction in (1), from what I understand, for the  “invariant-symmetric” property of

 $L^{IS}_{G_2,G_1}$

you need to assume that $G_1$ acts trivially on $X_1$, namely, $gx=x$, and for the “invariant-symmetric” property of $L^{IS}_{G_1,G_2}$ you need to assume that $G_2$ acts trivially on $X_2$, namely, $gy=y$. This means that your construction cannot describe any structure but the trivial symmetry. The proof of Theorem 1 in the appendix also shows that this is what you construct. Hence, the whole construction is vacuous.

What am I missing?

If I misinterpreted the construction, please explain. I will be happy to change my score.

**Questions:**

Some other problems that I found in the paper before realizing that the model (1) is problematic:

The term “independent groups”, for example at the bottom of page 1, should be replaced by commuting subgroups, or by “all inputs are acted upon independently by separate groups”.

Page 2: “invariant-symmetric”: Note that you simply define the identity action of $G_2$ on  the range of $f$ in $Y$. Namely, for any $y=f(x)\in Y$, you assume that $g_2y=y$. In what sense is this an interesting model? It describes no symmetry.

Section 3.1 Multiple Inputs - you mean that the sequence $(X_1,\ldots,X_N)$ is the input, not that this is a set of inputs. You should also formulate the setting as follows: the direct product group $G=(G_1,\ldots,G_N)$ acts on the input space $\mathbb{R}^{d_1,\ldots,d_N}$.

Large product groups : ``the subgroups $g_i$ act in the same order.’’ The order does not matter since $G$ is a direct product group. The subgroups $G_i$ commute.

“ whereas for constructing G-invariant models we do not need commutativity” It is not a matter of need. Since the subgroups $G_i$ commute by definition of direct product of groups, and by definition of group action, the action of the subgroups $G_i$ must commute.

Equation (1): what is $L^{IS}_{G_2,G_1}$

?
 You did not define it. You also did not define $L_{G_1}^{Eq}$ and $L_{G_2}^{Eq}$. There is a problem with this construction as I wrote above.

At this point I have to admit that I stopped reading. If my assessment of (1) is correct, the paper should be rejected. If I misunderstood the construction I apologize.

---

> ### Author Response · Authors · 2023-11-13
> **Response to reviewer gLGd**
>
> We clarify the weaknesses and questions raised by the reviewer. With these clarifications, we sincerely hope the reviewer will kindly reconsider reading through our paper. We strongly believe the reviewer will find it a compelling next step in designing efficient model agnostic equivariant networks, following the recent emergence of works in this area [1-4].
>
> [1] Basu et al. "Equi-tuning: Group equivariant fine-tuning of pretrained models." AAAI 2023.
>
> [2] Basu et al. "Equivariant few-shot learning from pretrained models." NeurIPS 2023
>
> [3] Kim et al. "Learning probabilistic symmetrization for architecture agnostic equivariance." NeurIPS, 2023.
>
> [4] Mondal et al. "Equivariant Adaptation of Large Pre-Trained Models." NeurIPS 2023.
>
> **Reviewer:** "... the model is vacuous, and can only apply for the trivial symmetry."
>
> **Response:** Please note that this is a misunderstanding since we design group equivariant models (not invariant-symmetric) for product groups G1xG2 for any two groups G1 and G2. In Thm. 2, we prove that our model design for the linear case covers the entire linear equivariant function space corresponding to product groups. Hence, clearly, our model does not only apply to trivial symmetries. Further, the competitive performance of our model design while respecting all the required equivariances also implies its meaningfulness.
> Please find more details on the nature of the invariant-symmetric fusion layers in the response below.
>
> **Reviewer:** "In page 2, the notion of “invariant-symmetric” seems strange."
>
> **Response:** Please note that we do not invent the notion of invariance-symmtery, we simply discover that it is present in linear equivariant layers of product groups and give it a name. This observation/discovery helps us design simpler networks going beyond the linear framework.
>
> As we show in Thm. 2, it is naturally present in linear models of product groups. When a linear group equivariant model for the first design is constructed for multiple groups, this notion of invariance-symmetry naturally arises as a fusion layer between the different inputs. This is similar to different fusion layers with symmetries that arise in Deep Sets [5], and DSS [6]. Unlike Deep Sets and DSS, we do not stack up linear layers to construct our full model. Instead, we give a model-agnostic design, which goes beyond linear design and works even for pretrained models, leading to the methods discussed in Sec. 3.3, 3.4.
>
> [5] Manzil et al. "Deep sets", NeurIPS 2017
> [6] Maron et al. "On learning sets of symmetric elements", ICML 2020.
>
> **Reviewer:** "Note that you simply define the identity action ... just the trivial symmetry."
>
> **Response:**
>
> - a) Please note that Pg. 2 only provides the def of invariance-symmetry (IS) that we discover in the **fusion layer of the equivariant network corresponding to product groups**. Thus **IS only represents a subcomponent of our model design**, it does not define the symmetry we are aiming for, which is group equivariance to product groups.
>
> - b) Please note that the reviewer seems to have misunderstood the definition of IS and confused it with trivial symmetry.  The two are different concepts as described below. Also, note that we make no assumptions on the IS function; instead, we provide functions f that precisely satisfy the IS constraint.
>     - trivial symmetry (https://en.wikipedia.org/wiki/Symmetry_group) means the group action corresponds to a group with one single element that does not perform any transformation to the object of the action.
>   - A function f: X -> Y has the property of IS with respect to ordered groups (G1, G2) where G1 acts on X and G2 acts on Y, if f(g1 x) = x (invariance) and g2 f(x) = f(x) (symmetry) for all x in X, g1 \in G1, g2  \in G2. From what we can gather, the reviewer may be confusing the notion of symmetry in our work f(x) = g2 f(x) to trivial symmetry group. Trivial symmetry group means a group with a single element which when acts on an object leaves the object unchanged. On the other hand, our notion of symmetric function simply means any function that remains unchanged with respect to group actions on the output of the function. This is very closely related to the notion of invariance (we call a function invariant when the output remains unchanged with respect to transformations on the input). On the other hand, we call a function symmetric when it remains unchanged with respect to a group transformation applied to the output. Consider the following example. Let f1: R^2-> R^2, f2: R^2->R^2 be functions defined as f1(x, y) = (1, 1), f2(x, y) = (1, 1) if (x, y) in {(0, 0), (0, 1), (-1, 0), (0, -1)}, else (0, 0). Here f1 is invariant whereas f2 is symmetric, with respect to 90 degree rotations of the input about the origin. Finally, we reemphasize we did not invent this notion of IS it naturally arises in equivariant network design for product groups, which we discover, name, and exploit for efficient model designs.

---

> > ### Author Response · Authors · 2023-11-13
> > **Response to reviewer gLGd (contd.)**
> >
> > **Reviewer:** The term “independent groups”, for example at the bottom of page 1, should be replaced by commuting subgroups, or by “all inputs are acted upon independently by separate groups”.
> >
> > **Response:** We are happy to make this modification for better clarity of the readers.
> >
> > **Reviewer:** Large product groups : ``the subgroups $G_i$ act in the same order.’’ The order does not matter since G is a direct product group.
> >
> > **Response:** Please note that we did not mean direct product groups in our work. We apologize for not writing it more clearly, we meant the more general semi-direct products, where the product is not commutative. Please note that the group products used in our experiments D4 (product of reflection and 90 degree rotation) are also semi-direct product groups, which is very common in the equivariance literature. We will fix this confusion in the updated draft along with the concerns from other reviewers.
> >
> > **Reviewer:** Equation (1): what is $L_{(G_2, G_1)}^{IS}$? You did not define it. You also did not define $L_{G_1}^{Eq}$ and $L_{G_2}^{Eq}$. There is a problem with this construction as I wrote above.
> >
> >
> > **Response:** As defined on pg. 4, $L_{G}^{Eq}$ denotes a G-equivariant linear network. Please note that in this definition, we do not restrict the properties of G. Hence, G could be any group. Thus, analogously, $L_{G_1}^{Eq}$ and $L_{G_2}^{Eq}$ are $G_1$-equivariant and $G_2$-equivariant linear networks.
> > Similarly, in the definition of $L_{(G_1, G_2)}^{IS}$, which are linear ($G_1, G_2$)-invariant-symmetric networks, we do not restrict what $G_1$ and $G_2$ could be. Thus, $L_{(G_2, G_1)}^{IS}$ is analogously defined as a linear ($G_2, G_1$)-invariant-symmetric network.
> >
> > We will make sure to add this clarification in the updated draft along with other updates.

---

> ### Comment · Reviewer_gLGd · 2023-11-13
> **Response to authors**
>
> Thank you for you response.
>
> Let me see if I can rephrase things more accurately so we can agree.
>
> For $f$ to be IS, you assume that for every $y$ in the range of $f$, and every $g_2\in G_2$, you have $y=g_2f$.
>
> This means that the range of $f$ is a subset of the following set of points in $Y$: the set $S$ of all $y\in Y$ such that $g_2y=y$ for every $g_2\in G_2$.  So $G_2$ acts trivially on the range of $f$. Namely, the action of $G_2$ is the identity there, and the range of $f$ is called fixed under $G_2$. In $S$, the action of $G_2$ is the trivial action (this is what I called informally a trivial symmetry.).
>
> Do we agree now on that?
>
> **I do see now that it is meaningful to assume that the rage of $f$ is fixed under the action.** For example, under the reflection action on images (action of the group $\{-1,1\}$), the output of $f$ must be in the set of symmetric (or even) images.
>
>
> Regarding replacing direct product with semidirect product, the image application indeed has a semidirect product group structure.
>
> Regarding the other applications, I cannot localize in the paper the definitions of the semidirect product group and its action in these applications. Can you write these definitions explicitly so we see that these are indeed semidirect product groups?
>
>
> I apologize for not reading in detail the whole paper the first time. The inconsistencies in group theory (the fact that the theory was for direct product groups but you used non-commuting groups in applications), and the confusion about triviality of the action led me to immediately reject the paper.
>
> Now, I need to see the revised paper so I can re-evaluate it.

---

> > ### Author Response · Authors · 2023-11-13
> > **Response to reviewer gLGd**
> >
> > We thank the reviewer for their prompt response. We have uploaded a new version of our paper with updated notations of products clearly indicating semi-direct products.
> >
> > Yes, we agree on the current understanding of the IS layer described by the reviewer.
> >
> > We have updated the notation of products and specifically clarified that we are working with semi-direct products to avoid any confusion. Please note semi-direct products are required for the experiments as the reviewer also agrees. Further, we would like to clarify that the semi-direct products reduce to simple direct products for the applications on texts since the groups there are acting on disjoint sets.

---

> ### Comment · Reviewer_gLGd · 2023-11-14
> **Order of semidirect product**
>
> Thank you.
> Note that semidirect product is not symmetric. One subgroup is normal and the other is not in general. So, if you define a nested semidirect product you need to use parenthesis that indicate in which order you apply the semidirect products.
>
> For example, does $G=G_1 \rtimes G_2 \rtimes G_2$ mean $(G_1 \rtimes G_2) \rtimes G_3$, where $G_1 \rtimes G_2$ is normal in $G$, and $G_1$ is normal in $G_1 \rtimes G_2$, or does it mean $G_1 \rtimes (G_2 \rtimes G_3)$, where $G_1$ is normal in $G$ and $G_2$ is normal in $G_2 \rtimes G_3$?

---

> > ### Author Response · Authors · 2023-11-14
> > **Response to reviewer gLGd**
> >
> > parentheses updated, thanks

---

> > > ### Comment · Reviewer_gLGd · 2023-11-15
> > > **Review of the revised paper**
> > >
> > > **Motivation**
> > >
> > > It is not clear how important the problem of taking a trained model and symmetrizing it. Usually, geometric deep learning is motivated by reducing the number of training parameters before training, not after training, to reduce the model complexity. This point should be discussed in the paper. What is the advantage and disadvantage in symmetrizing the model before vs after training. I think that this should be one main point in the paper.
> > >
> > >
> > > **Inconsistencies in group theory**
> > >
> > >
> > > The proofs in this paper are specifically for subgroups of the symmetric group and their representations as permutations. However, the paper presents the theory as if it is for general group actions. See for example (3) and Lemma 1:
> > >
> > > In Equation (3): what do you mean by “permutation group corresponding to $g\in G$”? In Maron et al the group was a subgroup of the permutation group, and $P(g)$ was the matrix representation of the permutation. You work with general actions until now. If you want to use this results you need to restrict the analysis to permutation representation.  This is also trie for Lemma 1. Note the difference between group action and group representation, and between general representation and permutation.
> > >
> > > After equation (5), fix to “is invariant-symmetric with respect to (Gi, Gj).” You also did not introduce the notation $M$.
> > >
> > > Equation (6): this formula also shows that your construction should be restricted to group representations and not general group actions. Without linearity of the action WRT the vector space on which it is defined the construction would not work.
> > >
> > >
> > > Applications:
> > >
> > > Regarding the language model application. As I explained above, the theory should be restricted to permutation representations on vector spaces. The language model application was described using a group action on a set which is not a vector space. You should write a paragraph, perhaps in the appendix, that explains how the input and output of the model in this case are in vector spaces and how the action is a permutation representation.
> > >
> > >
> > > **Suggestion**
> > >
> > > You do not really need the group to be a nested semidirect product group for the construction to work, right?
> > > You can formulate the setting as follows. Write that G is a set-direct product of subgroups $G_1,\ldots,G_n$. Namely, as a set it is a direct product of sets, but not as a group. The subgroups need not commute. Then you can give two examples: group direct product and nested group semidirect product in any order of nesting. If you formulate the setting this way, take special care to stress that the direct product is in the set sense and not the group sense.

---

> > > > ### Author Response · Authors · 2023-11-22
> > > > **Response to revised review**
> > > >
> > > > We thank the reviewer for their revised review. We provide responses to the concerns raised below.
> > > >
> > > > **Reviewer:** Motivation...It is not clear how important the problem of taking a trained model and symmetrizing it. Usually, geometric deep learning is motivated by reducing the number of training parameters before training, not after training, to reduce the model complexity.
> > > >
> > > > **Response:** As mentioned in the introduction of our paper, this work takes inspiration from previous works that leverage the benefits of equivariance as well as pretrained models [1-4].
> > > >
> > > >
> > > > Please note that making pretrained models equivariant is of crucial importance with the rise of large pretrained models. These large models are usually not trained with equivariances required for certain downstream tasks, but training them from scratch with group equivariance is extremely expensive and hence often not an option. Yet, using these large foundation models in downstream tasks shows excellent performance. Thus, the motivation of this work (focused on making pretrained model equivariant) is of great importance. Indeed, this research direction has gained recent importance as evident from several papers [1-4] in this direction. These are clearly mainstream papers on group equivariance published in respected conferences (NeurIPS and AAAI).
> > > >
> > > >
> > > > [1] Basu et al. "Equi-tuning: Group equivariant fine-tuning of pretrained models." AAAI 2023.
> > > >
> > > > [2] Basu et al. "Equivariant few-shot learning from pretrained models." NeurIPS 2023
> > > >
> > > > [3] Kim et al. "Learning probabilistic symmetrization for architecture agnostic equivariance." NeurIPS, 2023.
> > > >
> > > > [4] Mondal et al. "Equivariant Adaptation of Large Pre-Trained Models." NeurIPS 2023.
> > > >
> > > > **Reviewer:** Inconsistencies in group theory
> > > > The proofs in this paper are specifically for subgroups of the symmetric group and their representations as permutations. However, the paper presents the theory as if it is for general group actions. See for example (3) and Lemma 1:
> > > > In Equation (3): what do you mean by “permutation group corresponding to g \in G”? In Maron et al the group was a subgroup of the permutation group, and P(g) was the matrix representation of the permutation. You work with general actions until now. If you want to use this results you need to restrict the analysis to permutation representation. This is also trie for Lemma 1. Note the difference between group action and group representation, and between general representation and permutation.
> > > >
> > > > **Response:** We have now updated the statement in the paper to clarify that $G$ is a subgroup of a permutation group and let P(g) represent the group element corresponding to g \in G. We hope this clarifies the confusion.
> > > >
> > > >
> > > > Note that only Sec. 3.2 assumes the linearity of group action and also that of the model. From beyond Sec. 3.3, we carry the intuitions gained from 3.2 and design non-linear models for arbitrary group actions (not necessarily linear).
> > > >
> > > > **Reviewer:** After equation (5), fix to “is invariant-symmetric with respect to (Gi, Gj).” You also did not introduce the notation M.
> > > >
> > > > **Response:** We have updated the statement. We have now updated that M is a pretrained model in that equation.
> > > >
> > > > **Reviewer:** Equation (6): this formula also shows that your construction should be restricted to group representations and not general group actions. Without linearity of the action WRT the vector space on which it is defined the construction would not work.
> > > >
> > > > **Response:** We have now updated the notation in equation 6 from summation to concatenation, hence, removing any necessity for linearity for our construction to work. This works since our construction beyond Sec. 3.2 does not depend on any linearity assumption. We inadvertently had the summation notation in equation 6 before as it was used in the language experiments, where the output group action is indeed a permutation group action.
> > > >
> > > > **Reviewer:** Applications:
> > > > Regarding the language model application. As I explained above, the theory should be restricted to permutation representations on vector spaces. The language model application was described using a group action on a set which is not a vector space. You should write a paragraph, perhaps in the appendix, that explains how the input and output of the model in this case are in vector spaces and how the action is a permutation representation.
> > > >
> > > > **Response:** As discussed above, the updated notations do not depend on linearity anymore.
> > > >
> > > >
> > > > **Reviewer:** Suggestion
> > > > You do not really need the group to be a nested semidirect product group for the construction to work, right? You can formulate the setting as follows. ...take special care to stress that the direct product is in the set sense and not the group sense.
> > > >
> > > > **Response:** We thank the reviewer for the good suggestion. For now, we are keeping the semi-direct product notation in the draft, but will take this up when we are able.

---

### Official Review · Reviewer_ePZ4 · 2023-11-07

**Soundness:** 2 fair
**Presentation:** 2 fair
**Contribution:** 1 poor
**Rating:** 3
**Confidence:** 4

**Summary:**

This paper considers the problem of constructing equivariant networks in a model-agnostic manner. In particular, this paper tackles the setting in which the group can be decomposed as a product group. This leads to two principal tasks: 1.) the multi-input setting in which each group in the product acts on its respect input and 2.) the single input setting but the symmetry on this data type has a product structure. For the first setting, the authors propose IS fusion layer and characterize the entire space of linear equivariant functions with multiple inputs. They then extend this to the non-linear setting and prove universal approximation capabilities. Experiments are conducted in both settings and include multi-image classification and downstream applications of compositional generalization and language. The proposed approach is sometimes competitive with previous approaches but has the benefit of linear computational complexity w.r.t. to the number of groups.

**Strengths:**

The paper has a few strengths that I would like to highlight. First, the paper builds upon two recent papers by Basu et. al 2023 and Kim et. al 2023. and this allows the paper to lean on existing methodology. Thus the overall idea is relatively straightforward to understand. Moreover, the presented theory can be understood equally easily as it largely follows from the author's definitions and results in Maron et. al 2020.

**Weaknesses:**

Despite the stated strengths above; I have strong reservations regarding this paper. The first one is on the motivation. It is unclear to me why we would want to make an existing trained model equivariant. This is certainly the assumption in the downstream language experiments, but this is not at all a convincing demonstration. The task is contrived and does not fit in the broader equivariant literature.

**Large discrete product groups?**

In addition, to the lack of coherent motivation, I also found the claims in the paper to be unsupported. A large theme of this work is on building equivariance across **large discrete** product groups. In the multi-input experiments (Table 1) you have $N=4$ and the $C_4$ group. This is not a large product or large individual group. Similarly, in the compositional generalization experiment you write "The product groups are made of three smaller each of size two, and the largest product group considered is of size eight". Again this is not a large product group. A similar criticism can be attributed to the intersectional fairness experiments. So I find the entire claim and motivation for doing this work lacking. In fact, I would argue that you can just as easily do frame averaging and canonicalization in these settings. Thus at **minimum** these should be baselines.

**Experimental design and results**
The entire choice of experiments in this paper leans heavily on Basu et. al 2023's results. But I don't think the authors thought that the experimental setup in that paper might also be problematic in terms of highlighting the claims and goals. First, there is really no standard equivariant benchmark from many of the seminal equivariant papers. For example, you could have considered molecular datasets where you have $S_n$ and $SE(3)$. $S_n$ would be a much larger discrete group and you could consider discrete subgroups of $SE(3)$. Given the large literature of equivariant models in this space, the lack of this benchmark is alarming. In addition, if you really want to show discrete product group structure then there is a large body of work on latent space disentanglement via Linear symmetries which started from the seminal work of (Higgins et. al 2018). Completely ignoring this line of work and its benchmarks which are almost tailor-made to the setting considered in this work is questionable. Finally, I find the choice of compositional generalization and intersectional fairness via group theoretic notions quite a contrived task that runs counter to the actual practical goal of this work which is to scale up equivariant models for product groups.

With regard to results, there are many areas of improvement. To start off, the proposed approach does worse than equitune in some experiments (e.g. Fig 2). The authors do not have a convincing argument on why this is acceptable outside that their proposed approach has better-scaling properties w.r.t. to the number of groups in the product. Unfortunately, as I stated above this is not a large product so this is weird. Secondly, obvious baselines are missing. These include having an actual equivariant (architecture) model that is trained (not fine-tuned) post hoc in this manner. Also, frame averaging and canonicalization should be included.

**Questions:**

Please consider adding the following experiments.

1.) An experiment where $S_n$ is in the product. This can be a molecular task or not, but the equivariant literature has many examples of benchmarks.

2.) Adding an experiment with latent product symmetry. This would be a really convincing and better experiment in your setting.

3.) Can you please add the baselines mentioned in the weakness section?

4.) Can you please highlight the computational cost (iters/sec, flops, training time, inference time, etc...) of your approach versus frame averaging for your tasks. My guess is that it is quite similar given how small the group is.

---

> ### Author Response · Authors · 2023-11-22
> **Response to reviewer ePZ4**
>
> We thank the reviewer for their detailed feedback to improve the contributions of the paper. We aim to clarify the concerns raised by the reviewer.
>
> **Reviewer:** The first one is on the motivation. It is unclear to me why we would want to make an existing trained model equivariant.
>
> **Response:** Please note that making pretrained models equivariant is of crucial importance with the rise of large pretrained models. These large models are usually not trained with equivariances required for certain downstream tasks, but training them from scratch with group equivariance is extremely expensive and hence not often an option. Yet, using these large foundation models in downstream tasks shows excellent performance. Thus, the motivation of this work (focused on making pretrained model equivariant) is of great importance. Indeed, this research direction has gained recent importance as evident from several papers [1-4] in this direction. These are clearly mainstream papers on group equivariance published in respected conferences (NeurIPS and AAAI). Hence, we respectfully disagree that the considered task does not fit in the broader equivariant literature.
>
> Further, we do not believe that the language task is contrived. The problem of addressing social biases in generated text from language models and the used framework [5] is widely studied. Further, addressing the issue of bias in natural language generation is not trivial, as shown in several works such as [6]. Taking inspiration from the formulation of [1] that shows that equivariance can provably debias natural language with respect to defined groups, we aim to extend these equivariance techniques to larger social groups much more efficiently to help safer deployments of language models. One can further note that intersectionality has been a pernicious problem in debiasing, and our mathematical framework is very natural therein.
>
> [1] Basu et al. "Equi-tuning: Group equivariant fine-tuning of pretrained models." AAAI 2023.
>
> [2] Basu et al. "Equivariant few-shot learning from pretrained models." NeurIPS 2023
>
> [3] Kim et al. "Learning probabilistic symmetrization for architecture agnostic equivariance." NeurIPS, 2023.
>
> [4] Mondal et al. "Equivariant Adaptation of Large Pre-Trained Models." NeurIPS 2023.
>
> [5] Sheng et al. "The Woman Worked as a Babysitter: On Biases in Language Generation." EMNLP-IJCNLP 2019
>
> [6] Steed et al. "Upstream Mitigation Is Not All You Need: Testing the Bias Transfer Hypothesis in Pre-Trained Language Models." ACL 2022
>
> **Reviewer:** Large discrete product groups?
> In addition, to the lack of coherent motivation, I also found the claims in the paper to be unsupported. A large theme of this work is on building equivariance across large discrete product groups. In the multi-input experiments (Table 1) you have N=4 and the C_4 group. This is not a large product or large individual group. Similarly, in the compositional generalization experiment ... largest product group considered is of size eight". Again this is not a large product group. A similar criticism can be attributed to the intersectional fairness experiments. So I find the entire claim and motivation for doing this work lacking.
>
> **Response:** Our main contribution is intended to be theoretical, where the experiments are given to validate our theoretical claims. Hence, we considered products of small groups rather than large discrete product groups, and show that the gap in computational complexity (in terms of memory consumption) is as comes from our theory. We are happy to rewrite the experiments section to indicate that the experiments are meant to validate the theoretical findings and that we did not work with very large product groups since that extra computational cost was unnecessary for theory validation.

---

> > ### Author Response · Authors · 2023-11-22
> > **(Contd.) Response to reviewer ePZ4**
> >
> > **Reviewer:** In fact, I would argue that you can just as easily do frame averaging and canonicalization in these settings. Thus at minimum these should be baselines.
> >
> > **Response:** Please note that frame averaging is a general framework that encompasses equituning [1], lambda-equitune [2], probabilistic symmetrization [3], canonicalization [7], and equivariant adaptation [4].
> >
> >
> > We include comparison to a frame averaging method, equituning, which is known to perform well with pretrained models. The canonicalization paper, on the other hand, focuses on model-agnostic equivariance but does not explore the effect of its use on pretrained models. Later, [4], which appeared just a few days before the ICLR deadline, showed that naive canonicalization is not good enough for use with pretrained models, and hence modified/improved versions are required. Since this work appeared very close to the ICLR deadline, we are unable to provide detailed comparisons.
> >
> >
> > Additionally, note that canonicalization is non-trivial for translation tasks such as SCAN. This is because performing canonicalization [7] for a translation task with M input tokens and N output tokens would require finding the group element corresponding to the entire sequence of input and applying the same group element on the entire output sequence. This is not possible due to different numbers of input and output tokens. However, it is still possible to apply equitune and multi-equitune in this case since the group action is defined on the vocabulary space of the tokens and applied to each token simultaneously. Applying the group action simultaneously on each of the tokens is not possible for canonicalization because the group element is dependent on the entire input and output sequences.
> >
> >
> > Further, note that our work is complementary to the methods of [2-4]. Thus, the advantages obtained by [2, 3] over [1] and [4] over [7] can also be obtained by applying them to our algorithm. To keep our exposition simple, we start with equitune and show how it becomes expensive for large groups and hence using group decomposition can massively improve time complexity at the cost of a small drop in performance. Note that as the pretrained models get larger, the drop in performance decreases (cf. Fig. 4), while the improvement in complexity remains the same.
> > [7] Kaba et a. "Equivariance with Learned Canonicalization Functions" ICML 2023
> >
> > **Reviewer:** Experimental design and results The entire choice of experiments in this paper leans heavily on Basu et. al 2023's results. But I don't think the authors thought that the experimental setup in that paper might also be problematic in terms of highlighting the claims and goals.
> >
> > **Response:** Please note that our intent in designing experiments is to validate the advantages gained from our algorithm as compared to existing model-agnostic frame-averaging-based methods such as equitune. Clearly, the theoretical advantage is huge (sum of group sizes vs. product of group sizes) and is validated in our experiments.
> >
> > It is unclear to us how the experimental setup in [2] is problematic and would appreciate further guidance from the reviewer. We believe that the setup in [2] perfectly fits our framework and allows us to demonstrate the advantage gained by using our method on product groups used in the setup of [2].
> >
> > **Reviewer:** First, there is really no standard equivariant benchmark from many of the seminal equivariant papers ... the lack of this benchmark is alarming.
> >
> > **Response:** Response:
> > Please note that works such as [8] considered a similar problem of constructing approximately equivariant networks for larger groups using networks that are equivariant to smaller groups. Our work, just like [8], has mostly focused on images (and additionally, text).
> >
> >
> > We believe the reviewer is suggesting an SE(3)xS_n experiment in the context of our second formulation where one single model has group action applied that is a product of multiple smaller groups. However, we are unclear as to how this experiment would elucidate properties of our method. Our method is useful when we have large non-equivariant pretrained models and we want good finetuning or zero-shot performance. For example, if the zero-shot performance requires multiple passes through the pretrained model, our method shows how to reduce the number of forward passes while showing excellent equivariant performance.  We are happy to receive specific guidance from the reviewer on molecular experiments (e.g. with non-equivariant pretrained models) that would provide insights into our method.
> >
> > [8] Maile et al. "Equivariance-aware Architectural Optimization of Neural Networks." ICLR 2023

---

> > > ### Author Response · Authors · 2023-11-22
> > > **(contd.) Response to reviewer ePZ4**
> > >
> > > **Reviewer:** ...to show discrete product group structure then there is a large body of work on latent space disentanglement via Linear symmetries which started from the seminal work of (Higgins et. al 2018)...
> > >
> > > **Response:** Unfortunately, again, it is unclear to us how these suggested experiments fit with the problem we pursue. We would again kindly request the reviewer to provide further detailed guidance on suggested experiments.
> > >
> > > **Reviewer:** ... choice of compositional generalization and intersectional fairness via group theoretic notions quite a contrived ...
> > >
> > > **Response:** We would reiterate that the chosen tasks of solving compositional generalization and intersectional fairness are important.
> > >
> > >
> > > We would point the reviewer to a large literature on solving compositional generalization using group equivariance and data augmentation, e.g. [9-12], which use the SCAN dataset as a standard dataset to check compositional generalization.
> > >
> > >
> > > Similarly, our framework for intersectional fairness and its evaluation is fully based on the popular work of Sheng et al. [12].
> > >
> > >
> > > Moreover, we think these experiments perfectly fit the main goal of the paper: to provide an efficient method for performing equivariant finetuning and zero-shot evaluation from pretrained models. In fact, our entire theory was developed to tackle these problems: one of our main goals is to make large pretrained models fair and unbiased. Thus, showing that these fairness and compositional generalization tasks can be performed more efficiently with negligible drop in performance shows that these models can be more easily deployed in practice while guaranteeing their fairness and compositional generalization.
> > >
> > >
> > > [9] Gordon et al. "Permutation equivariant models for compositional generalization in language."  ICLR 2019
> > >
> > > [10] Zhao et al. "Toward Compositional Generalization in Object-Oriented World Modeling." ICML 2022.
> > >
> > > [11] White, and Cotterell. "Equivariant Transduction through Invariant Alignment." Proceedings of the 29th International Conference on Computational Linguistics. 2022.
> > >
> > > **Reviewer:** With regard to results, ... the proposed approach does worse than equitune in some experiments (e.g. Fig 2). The authors do not have a convincing argument on why this is acceptable ...
> > >
> > > **Response:** Our method is designed so that we obtain better much computational complexity at the expense of negligible drop in accuracy. The gained computational benefit (sum of group sizes vs. product of group sizes) is huge. Note that the improvement in computational complexity comes with a small drop in accuracy, which reduces with increasing model size (cf. Fig 4C).
> > >
> > >
> > > The drop in accuracy occurs for the following reason. Consider G to be a product of two smaller groups. Then equitune corresponds to a double summation of symmetrization over a pretrained model. On the other hand, we avoid this double summation by adding some invariance to the input and symmetry in the output in equation 6. This invariance in the input causes small drops in the accuracy. But, we believe that the gained computational complexity overshadows the negligible drop in performance that decreases with increase in model size (as seen from Fig. 4). The drop in accuracy is now explained in detail in the new Sec. G in the appendix.
> > >
> > > **Reviewer:** Secondly, obvious baselines are missing. These include having an actual equivariant (architecture) model that is trained (not fine-tuned) post hoc in this manner. Also, frame averaging and canonicalization should be included.
> > >
> > > **Response:** It is confusing to us how we should add models trained for several experiments such as GPT2 or CLIP which are large pretrained models. The purpose of the work is to leverage the many advantages of pretrained models. Hence, we are not sure how training equivariant networks from scratch is a suitable baseline. Rather, the baseline, as we have already considered, should be other model-agnostic methods used with pretrained models.
> > >
> > > **Reviewer:** Can you please highlight the computational cost (iters/sec, flops, training time, inference time, etc...) of your approach versus frame averaging for your tasks. My guess is that it is quite similar given how small the group is.
> > > Response: We provide comparison of our method (multi-equitune) with frame-averaging-based methods such as equitune in Tab. 7 and 8 in the appendix.
> > >
> > > **Response:** We provide comparison of our method (multi-equitune) with frame-averaging-based methods such as equitune in Tab. 7 and 8 in the appendix.

---

> > ### Comment · Reviewer_ePZ4 · 2023-11-22
> > **Re: Part 1 of Response**
> >
> > Thank you for responding to my review. I still disagree with your motivations. I believe the problem domains that you tackle are individually important (e.g. fairness, compositional generalization, equivariance), but the intersection of equivariance + X in this paper is not motivated / interesting enough. I have read the cited papers you mention in your response and I still maintain my position that---very respectfully---the language task you consider is very toy and contrived.

---

> ### Comment · Reviewer_ePZ4 · 2023-11-22
> **Re: Response part 2**
>
> Thank you again for the second part of your response. I remain unconvinced as before and I will maintain my rating. I encourage the authors to think about my initial rebuttal---especially the experimental design---more carefully. I think the authors would greatly benefit from departing and leaning on the line of work related to Equitune.

---

> > ### Author Response · Authors · 2023-11-22
> > **Response to Re: Response part 2**
> >
> > We thank the reviewer for their prompt response.
> >
> > We did think through several setups of experiments including QM9 for molecules, and latent space product symmetries. However, we believe these experiments do not directly fit our framework. E.g., for the case of molecules with S_n x SO(2) symmetry, it is not clear if separating out S_n and SO(2) symmetries will lead to expressive neural networks.
> >
> > We think the best impact of the work is in the case of discrete product groups that are not parametrizable (hence, most methods designed for large groups such as canonicalization do not easily apply), e.g., our language-based experiments.
> >
> > We would highly appreciate it if the reviewer could kindly elaborate on their suggestion for suitable experiments to improve the impact of our work.

---

### Meta-Review · Area_Chair_RnYj · 2023-12-05

**Metareview:**

This paper propose a model-agnostic algorithm to build equivariant networks in the case of a group action that can be decomposed as a group product. The reviewers acknowledged the relevance an importance of this research direction. The reviewers noted several weaknesses for this paper:
- The motivation and potential impact of the idea: it is not clear how the specific contribution of the paper (making equivariant pretrained models) would have an impact. More especially such a contribution could be useful if it could scale to large networks and large input (and thus groups) which is the second main concern.
- The experiments could be improved to consider larger groups.
- After several back and forth with the reviewers it is clear that the theory was lacking precision.

I would suggest the author to rethink their story. In particular if they want to put the emphasis on the fact that their method is useful for large groups they should provide experiments with large groups.

**Justification For Why Not Higher Score:**

Clear consensus by the reviewers.

**Justification For Why Not Lower Score:**

NA

---

### Decision · Program_Chairs · 2024-01-16

Reject